# Curve Your Enthusiasm: Concurvity Regularization in Differentiable Generalized Additive Models

**Julien Siems**[*†‡]
University of Freiburg
siemsj@cs.uni-freiburg.de

**Konstantin Ditschuneit**[*‡]
Scenarium AI
ko.ditschuneit@gmail.com

**Winfried Ripken**[*]
Merantix Momentum
winfried.ripken@merantix.com

**Alma Lindborg**[*]
Merantix Momentum
alma.lindborg@merantix.com

**Maximilian Schambach**
Merantix Momentum
maximilian.schambach@merantix.com

**Johannes S. Otterbach**[‡]
nyonic
johannes@nyonic.ai

**Martin Genzel**
Merantix Momentum
martin.genzel@merantix.com

## Abstract

Generalized Additive Models (GAMs) have recently experienced a resurgence in popularity due to their interpretability, which arises from expressing the target value as a sum of non-linear transformations of the features. Despite the current enthusiasm for GAMs, their susceptibility to concurvity – i.e., (possibly non-linear) dependencies between the features – has hitherto been largely overlooked. Here, we demonstrate how concurvity can severely impair the interpretability of GAMs and propose a remedy: a conceptually simple, yet effective regularizer which penalizes pairwise correlations of the non-linearly transformed feature variables. This procedure is applicable to any differentiable additive model, such as Neural Additive Models or NeuralProphet, and enhances interpretability by eliminating ambiguities due to self-canceling feature contributions. We validate the effectiveness of our regularizer in experiments on synthetic as well as real-world datasets for time series and tabular data. Our experiments show that concurvity in GAMs can be reduced without significantly compromising their prediction quality, improving interpretability and reducing variance in the feature importances. [1]

## 1 Introduction

Interpretability has emerged as a critical requirement of machine learning models in safety-critical decision-making processes and applications subject to regulatory scrutiny. Its importance is particularly underscored by the need to ensure fairness, unbiasedness, accountability and transparency in

---

[*]Equal contribution

[†]Corresponding author

[‡]Work done while at Merantix Momentum

[1]Code: https://github.com/merantix-momentum/concurvity-regularization

37th Conference on Neural Information Processing Systems (NeurIPS 2023).

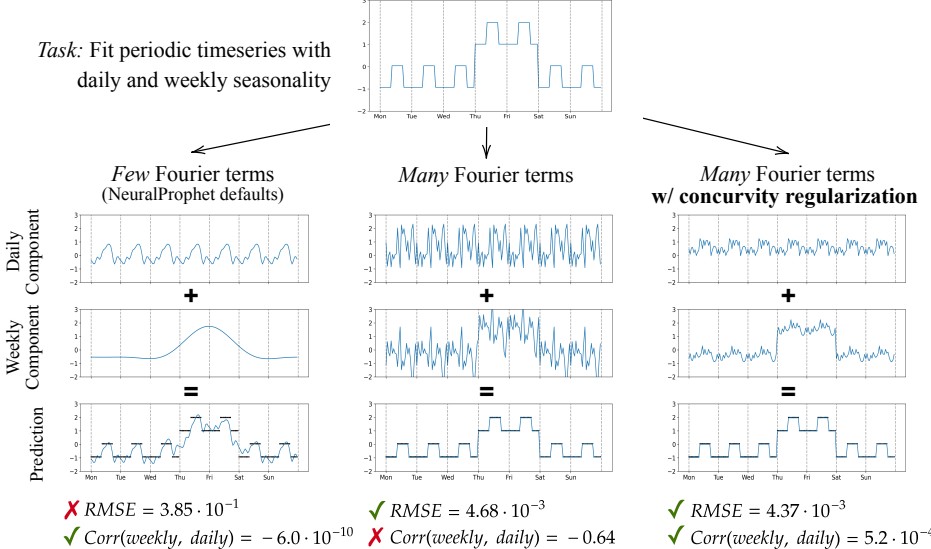

Figure 1: Concurvity in a NeuralProphet model: Fitting a time series composed of daily and weekly seasonalities, each represented by Fourier terms. (left) Using few Fourier terms results in uncorrelated components but a poor fit. (middle) A more complex model improves the fit but sacrifices interpretability due to self-canceling high-frequency terms. (right) The same complex model, but with our *regularizer*, achieves both good predictive performance and interpretable (decorrelated) components. See Section 4.2 for more details.

many applications and domains [6, 42], such as medical diagnoses [10], loan approvals [4], and hiring practices [16]. In such cases, model interpretability can even be favored over prediction accuracy [31].

A popular model class for interpretable machine learning is *Generalized Additive Models* (*GAMs*) [24], in which the target variable is expressed as a sum of non-linearly transformed features. GAMs combine the interpretability of (generalized) linear models with the flexibility of capturing non-linear dependencies between the features and the target. GAMs have recently seen a resurgence in interest with prominent examples being *Neural Additive Models* (*NAMs*) [2] and its variants [11, 19, 20, 39, 50] for tabular data, as well as *Prophet* [45] and *NeuralProphet* [46] for time-series forecasting. Both domains will be further explored in our experiments.

A significant obstacle to the interpretability of additive models is the phenomenon of *concurvity* [9]. As a non-linear analog to multicollinearity, concurvity refers to the presence of strong correlations among the non-linearly transformed feature variables. Similarly to multicollinearity, concurvity can impair interpretability because parameter estimates become unstable when features are correlated [40], resulting in highly disparate interpretations of the data depending on the model initialization. Although this issue is known and addressed by various techniques such as variable selection [28] in traditional GAMs, it has been overlooked in more recent works. Unlike the prevalent GAM package *mgcv* [47], we are not aware of any differentiable GAM implementations that include concurvity metrics.

In this work, we propose a novel regularizer for reducing concurvity in GAMs by penalizing pairwise correlations of the non-linearly transformed features. Reducing concurvity improves interpretability because it promotes the isolation of feature contributions to the target by eliminating potentially correlated or self-canceling transformed feature combinations. As a result, the model becomes easier to inspect by clearly separating individual feature contributions. In addition, our regularizer encourages the model to learn more consistent feature importances across model initializations, which increases interpretability. The trade-off between increased interpretability and prediction quality will be further explored in Section 4.

Figure 1 provides a first intuition of the practical effects of concurvity and how they can be addressed by our regularizer. We use the additive time-series model NeuralProphet [46] in this example, restricted to daily and weekly seasonality components. Here, each component is modeled via periodic functions with adjustable complexity. For more details on the experiment, see Section 4.2. We find that while the default NeuralProphet parameters effectively mitigate concurvity by producing a very

simple model, they provide a worse fit to the data than the more complex models. However, if left unregularized, a more complex model is subject to strong correlation between the seasonalities, an effect visually apparent in self-canceling periodic components in the middle column of Figure 1. In contrast, when using our regularization, the seasonalities are less correlated, resulting in a clearer separation between the components. While the predictive performance of the two complex models is comparable, the regularized model is more interpretable because daily and weekly effects are clearly separated.

This idea forms the basis of our argument: Comprehending a standard GAM typically requires answers to two questions: How do the features relate to my output? How do the features interact, possibly negating their contributions? By applying concurvity regularization, we simplify the interpretation of a GAM by eliminating the need for the second question.

Our main contributions can be summarized as follows:

1. We showcase the susceptibility of modern GAMs to concurvity and present a revised formal definition of the concept.
2. We propose a concurvity regularizer applicable to any differentiable GAM.
3. We validate our approach experimentally on both synthetic and real-world data, investigating the trade-off between concurvity and prediction quality, as well as the impact of regularization on interpretability.

## 2 Background

### 2.1 Generalized Additive Models

*Generalized Additive Models* (*GAMs*) [24] form a class of statistical models that extends Generalized Linear Models [36] by incorporating non-linear transformations of each feature. Following [24], GAMs can be expressed as:

$$g\big(\mathbb{E}\big[Y|X\big]\big) = \beta + \sum\nolimits_{i=1}^{p} f_i(X_i)\,, \tag{GAM}$$

where $Y = (y_1, \ldots, y_N) \in \mathbb{R}^N$ is a vector of $N$ observed values of a target (random) variable, $X = [X_1, \ldots, X_p] \in \mathbb{R}^{N \times p}$ assembles the observed feature variables $X_i = (x_{i,1}, \ldots, x_{i,N}) \in \mathbb{R}^N$, and $f_i : \mathbb{R} \to \mathbb{R}$ are univariate, continuous *shape functions* modeling the individual feature transformations.[4] Furthermore, $\beta \in \mathbb{R}$ is a learnable global offset and $g : \mathbb{R} \to \mathbb{R}$ is the link function that relates the (expected) target value to the feature variables, e.g., the logit function in binary classification or the identity function in linear regression. The shape functions $f_i$ precisely describe the contribution of each individual feature variable in GAMs, and can be visualized and interpreted similarly to coefficients in a linear model. This allows practitioners to fully understand the learned prediction rule and gain further insights into the underlying data.

While early GAMs primarily used splines [24] or boosted decision trees [10, 34, 35] to model $f_i$, more recent GAMs such as *Neural Additive Models* (*NAMs*) [2] use multilayer perceptrons (MLPs) to fit the functions $f_i$, benefiting from the universal approximation capacity of neural networks [15] as well as the support of automatic differentiation frameworks [1, 8, 38] and hardware acceleration. As a result, one can now solve the GAM fitting problem

$$\text{GAM-Fit:} \quad \min_{\beta,(f_1,\ldots,f_p)\in\mathcal{H}} \mathbb{E}\big[L\big(Y, \beta + \sum_{i=1}^{p} f_i(X_i)\big)\big] \tag{1}$$

by common deep learning optimization techniques such as mini-batch stochastic gradient descent (SGD). Here, $L : \mathbb{R} \times \mathbb{R} \to \mathbb{R}$ is a loss function and $\mathcal{H} \subset \{(f_1, \ldots, f_p) \mid f_i : \mathbb{R} \to \mathbb{R}\}$ any function class with differentiable parameters, e.g., MLPs or periodic functions like in NeuralProphet [46].

### 2.2 Multicollinearity and Concurvity

Multicollinearity refers to a situation in which two or more feature variables within a linear statistical model are strongly correlated. Formally, this reads as follows:[5]

---

[4]As usual, when $f_i$ or $g$ are applied to a vector, their effect is understood elementwise.

[5]Our definition is based on a fixed (deterministic) feature design, but one could also formulate a probabilistic version, see also Appendix A.2(1).

**Definition 2.1** (Multicollinearity). *Let $X_1, \ldots, X_p \in \mathbb{R}^N$ be a set of feature variables where $X_i = (x_{i,1}, \ldots, x_{i,N}) \in \mathbb{R}^N$ represents $N$ observed values. We say that $X_1, \ldots, X_p$ are (perfectly) multicollinear if there exist $c_0, c_1, \ldots, c_p \in \mathbb{R}$, not all zero, such that $c_0 + \sum_{i=1}^{p} c_i X_i = 0$.*

According to the above definition, every suitable linear combination of features that fits a target variable, say $Y \approx d_0 + \sum_{i=1}^{p} d_i X_i$, can be modified by adding a trivial linear combination $c_0 + \sum_{i=1}^{p} c_i X_i = 0$, i.e., $Y \approx (c_0 + d_0) + \sum_{i=1}^{p} (c_i + d_i) X_i$. So there exist other (possibly infinitely many) equivalent solutions with different coefficients. This can make individual effects of the features on a target variable difficult to disambiguate, impairing the interpretability of the fitted model. However, even in the absence of *perfect* multicollinearity, difficulties may arise.[6] For example, if two features are strongly correlated, estimating their individual contributions becomes challenging and highly sensitive to external noise. This typically results in inflated variance estimates for the linear regression coefficients [40], among other problems [17].

The notion of concurvity was originally introduced in the context of GAMs to extend multicollinearity to non-linear feature transformations [9]. In analogy with our definition of multicollinearity, we propose the following definition of concurvity:

**Definition 2.2** (Concurvity). *Let $X_1, \ldots, X_p \in \mathbb{R}^N$ be a set of feature variables and let $\mathcal{H} \subset \{(f_1, \ldots, f_p) \mid f_i : \mathbb{R} \to \mathbb{R}\}$ be a class of functions. We have (perfect) concurvity w.r.t. $X_1, \ldots, X_p$ and $\mathcal{H}$ if there exist $(g_1, \ldots, g_p) \in \mathcal{H}$ and $c_0 \in \mathbb{R}$ such that $c_0 + \sum_{i=1}^{p} g_i(X_i) = 0$ with $c_0, g_1(X_1), \ldots, g_N(X_N)$ not all zero.*

Technically, concurvity simply amounts to the collinearity of the transformed feature variables, and Definition 2.1 is recovered when $\mathcal{H}$ is restricted to affine linear functions. Concurvity poses analogous challenges to multicollinearity: *Any* non-trivial zero-combination of features can be added to a solution of GAM-Fit in Equation (1), rendering the fitted model less interpretable as each feature's contribution to the target is not immediately apparent. Finally we note that, although concurvity can be considered as a generalization of multicollinearity, neither implies the other in general, see Appendix A.2(2) for more details.

## 3 Concurvity Regularizer

Concurvity easily arises in highly expressive GAMs, such as NAMs, since the mutual relationships between the functions $f_i$ are not constrained while solving GAM-Fit in Equation (1). This results in a large, degenerate search space with possibly infinitely many equivalent solutions. To remedy this problem, it appears natural to constrain the function space $\mathcal{H}$ of (1) such that the shape functions $f_i$ do not exhibit spurious mutual dependencies. Here, our key insight is that *pairwise uncorrelatedness is sufficient to rule out concurvity*. Indeed, using $\mathcal{H}$ from Definition 2.2, let us consider the subclass

$$\mathcal{H}_\perp := \big\{ (f_1, \ldots, f_p) \in \mathcal{H} \mid \mathrm{Corr}\big(f_i(X_i), f_j(X_j)\big) = 0, \ \forall\, i \neq j \big\} \subset \mathcal{H},$$

where $\mathrm{Corr}(\cdot, \cdot)$ is the Pearson correlation coefficient. It is not hard to see that concurvity w.r.t. $X_1, \ldots, X_p$ and $\mathcal{H}_\perp$ is impossible, regardless of the choice of $\mathcal{H}$ (see Appendix A.1 for a proof). From a geometric perspective, $\mathcal{H}_\perp$ imposes an orthogonality constraint on the feature vectors. The absence of concurvity follows from the fact that an orthogonal system of vectors is also linearly independent. However, it is not immediately apparent how to efficiently constrain the optimization domain of Equation (1) to $\mathcal{H}_\perp$. Therefore, we rephrase the above idea as an unconstrained optimization problem:

$$\text{GAM-Fit}_\perp: \quad \min_{\beta, (f_1, \ldots, f_p) \in \mathcal{H}} \mathbb{E}\Big[ L\big(Y, \beta + \textstyle\sum_{i=1}^{p} f_i(X_i)\big) \Big] + \lambda \cdot R_\perp(\{f_i\}_i, \{X_i\}_i), \qquad (2)$$

where $R_\perp : \mathcal{H} \times \mathbb{R}^{N \times p} \to [0, 1]$ denotes our proposed *concurvity regularizer*:

$$R_\perp\big(\{f_i\}_i, \{X_i\}_i\big) := \tfrac{1}{p(p-1)/2} \sum_{i=1}^{p} \sum_{j=i+1}^{p} \big| \mathrm{Corr}\big(f_i(X_i), f_j(X_j)\big) \big|.$$

Using the proposed regularizer, GAM-Fit$_\perp$ simultaneously minimizes the loss function and the distance to the decorrelation space $\mathcal{H}_\perp$. In situations where high accuracy and elimination of

---

[6]Informally, non-perfect multicollinearity describes situations where $\sum_{i=1}^{p} c_i X_i \approx 0$. However, a formal definition would also require an appropriate correlation or distance metric.

concurvity cannot be achieved simultaneously, a trade-off between the two objectives occurs, with the regularization parameter $\lambda \geq 0$ determining the relative importance of each objective. An empirical evaluation of this objective trade-off is presented in the subsequent experimental section.

Since $R_\perp$ is differentiable almost everywhere, GAM-Fit$_\perp$ in Equation (2) can be optimized with gradient descent and automatic differentiation. Additional computational costs arise from the quadratic scaling of $R_\perp$ in the number of additive components, although this can be efficiently addressed by parallelization (see Appendix E.3.2 for a runtime analysis). A notable difference to traditional regularizers like $\ell_1$- or $\ell_2$-penalties is the dependency of $R_\perp$ on the data $\{X_i\}_i$. As a consequence, the regularizer is also affected by the batch size and hence becomes more accurate with larger batches. It is also worth noting that our regularizer does not necessarily diminish the quality of the data fit, for instance, when the input feature variables are stochastically independent, see Appendix A.2(3).

Common spline-based concurvity metrics proposed in the literature [28, 47] are not directly applicable to other differentiable models such as NAMs or NeuralProphet. Thus, similarly to [40], we decided to report the average $R_\perp(\{f_i\}_i, \{X_i\}_i) \in [0, 1]$ as a model-agnostic measure of concurvity.

# 4 Experimental Evaluation

In order to investigate the effectiveness of our proposed regularizer, we will conduct evaluations using both synthetic and real-world datasets, with a particular focus on the ubiquitous applications of GAMs: tabular and time-series data. For the experiments involving tabular data, we have chosen to use Neural Additive Models (NAMs), as they are differentiable and hence amenable to our regularizer. For time-series data, we investigate NeuralProphet models which contain an additive component modeling seasonality. Further elaboration on our experimental setup, including detailed specifications and parameters, can be found in Appendix C.

## 4.1 Toy Examples

In the following, we design and investigate two instructive toy examples to facilitate a deeper comprehension of the proposed regularizer as well as the relationship between the regularization strength $\lambda$ and the corresponding model accuracy. Results for an additional toy example from [28] are presented in Appendix E.2.

**Toy Example 1: Concurvity regularization with and without multicollinearity**    To compare the effect of concurvity regularization on model training in the presence of multicollinearity, we generate synthetic data according to the linear model $Y = 1 \cdot X_1 + 0 \cdot X_2$, where $X_1$ and $X_2$ are each drawn from a uniform distribution. We consider three different settings of input feature correlation: (1) the stochastically independent case where $X_1$ and $X_2$ are independently sampled, (2) the strongly correlated case where $\text{Corr}(X_1, X_2) = 0.9$, and (3) the perfectly correlated case with $X_1 = X_2$.

We first investigate the effect of concurvity regularization on the contribution of each feature to the target by measuring the correlation of the transformed features $f_1(X_1)$, $f_2(X_2)$ with the target variable $Y$. The results are shown in Figure 2a. In the stochastically independent case, the NAM accurately captures the relationship between the input features and target variable regardless of the regularization setting, as observed by the high correlation for $f_1(X_1)$ and zero correlation for $f_2(X_2)$ with the target. This result emphasizes the minimal impact of the regularizer when features are independent (see Appendix A.2(2) for details). In the perfectly correlated case, the NAM trained without concurvity regularization displays a high correlation for both $f_1(X_1)$ and $f_2(X_2)$ with the target, thus using both features for its predictions. In contrast, when concurvity regularization is applied, the NAM is pushed towards approximately orthogonal $f_i$, which encourages feature selection, as indicated by high correlation of either $f_1(X_1)$ or $f_2(X_2)$ with the target as there is no natural preference. Interestingly, the situation is slightly different in the strongly correlated case, as $X_1$ is more likely to be selected here, which means that the model has correctly identified the true predictive feature. This nicely illustrates the impact of the proposed regularization on correlated feature contributions. A more detailed visualization of the impact of regularization under perfectly correlated features can be found in Figure 8 in Appendix E.1.

Secondly, we examine the trade-off between the validation RMSE and concurvity $R_\perp$ in the perfectly correlated case in Figure 2b. Our findings suggest that with moderate regularization strengths $\lambda$, we can effectively eradicate concurvity without compromising the accuracy of the model. Only when

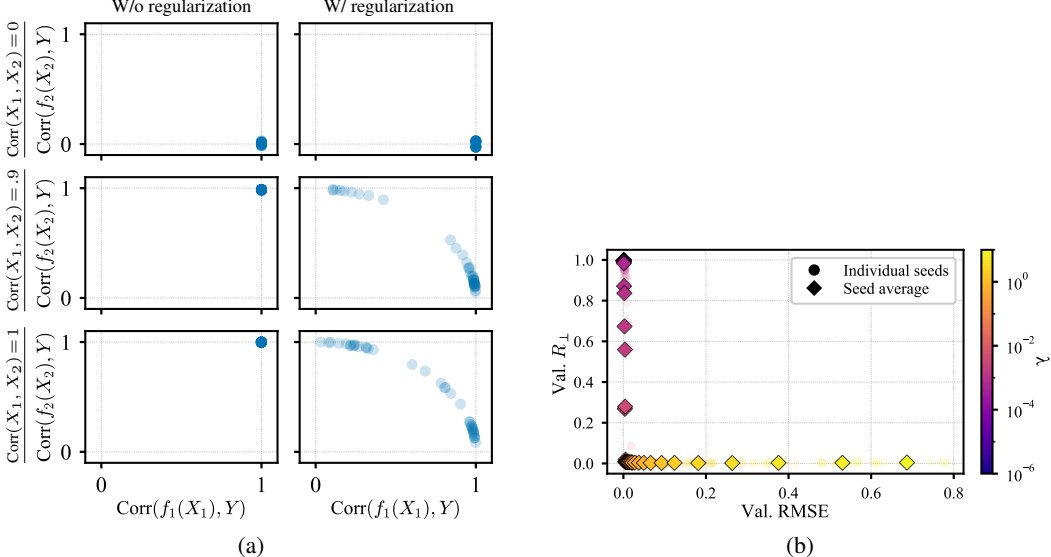

Figure 2: Results for Toy Example 1. (a) Effect of concurvity regularization on uncorrelated and correlated features. For each of the six settings, 40 random initializations were evaluated. (b) Trade-off curve between model accuracy (validation RMSE) and measured concurvity (validation $R_\perp$). Results are averaged over 10 random initializations per regularization strength $\lambda$.

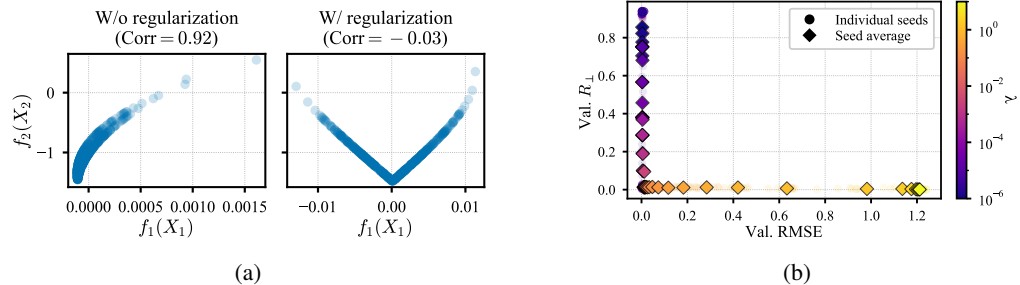

Figure 3: Results for Toy Example 2. (a) Comparison of transformed feature correlation with and without concurvity regularization. (b) Trade-off curve between model accuracy (validation RMSE) and measured concurvity ($R_\perp$).

the regularization strength is considerably high, the RMSE is adversely affected, without any further reduction in the measured concurvity.

**Toy Example 2: Concurvity regularization in the case of non-linearly dependent features** Next, we evaluate our regularizer in the case of non-linear relationships between features, a setting to which it is equally applicable. To this end, we design an experiment with feature variables that are uncorrelated, but not stochastically independent due to a non-linear relationship. We choose $X_1 = Z$ and $X_2 = |Z|$ where $Z$ is a standard Gaussian, and let $Y = X_2$ be the target variable. In this case, there is no multicollinearity by design, but the model may still learn perfectly correlated feature transformations. For example, a NAM could learn $f_1 = |\cdot|$ and $f_2 = \text{id}$, then $f_1(X_1) = f_2(X_2)$, which are fully correlated, yet providing a perfect fit. For our experiment, we use the same NAM model configuration as in the previous toy example.

The transformed features for the NAM fitted with and without regularization are visualized in Figure 3a. We find that the regularizer has effectively led the NAM to learn decorrelated feature transformations $f_1$ and $f_2$, reflected in the striking difference in feature correlation ($\text{Corr} = -0.03$ for the regularized NAM compared to $\text{Corr} = 0.92$ for the unregularized model). Moreover, these results suggest that the regularized model seems to have learned the relationship between the features, where $f_2(X_2)$ seems to approximate $|f_1(X_1)|$.

Finally, a trade-off curve of the validation RMSE and $R_\perp$ is shown in Figure 3b, illustrating that even in the case of non-linearly dependent features, our proposed regularizer effectively mitigates the measured concurvity $R_\perp$ with minimal impact on the model's accuracy as measured by the RMSE.

## 4.2 Time-Series Data

In this section, we provide more context and additional results for the motivational example in Figure 1 on time-series forecasting using NeuralProphet [46], which decomposes a time-series into various additive components such as seasonality or trend. In NeuralProphet, each seasonality $S_p$ is modeled using periodic functions as

$$S_p(t) = \sum_{j=1}^{k} a_j \cos\left(2\pi jt/p\right) + b_j \sin\left(2\pi jt/p\right)$$

where $k$ denotes the number of Fourier terms, $p$ the periodicity, and $a_j$, $b_j$ are the trainable parameters of the model. In the synthetic example of Figure 1, we restrict the NeuralProphet model to two components, namely a weekly and daily seasonality. Our overall reduced model is therefore given by $\hat{y}_t = S_{24\text{h}}(t) + S_{7\text{d}}(t)$.

If $k$ is sufficiently large, it can cause the frequency ranges of $S_{24\text{h}}$ and $S_{7\text{d}}$ to overlap, leading to concurvity in the model. The default values of $k$ provided by NeuralProphet for $S_{24\text{h}}$ ($k = 6$) and $S_{7\text{d}}$ ($k = 3$) are intentionally kept low to avert such frequency overlap, as demonstrated in Figure 1 (left). However, this "safety measure" comes at the cost of prediction accuracy due to reduced model complexity.

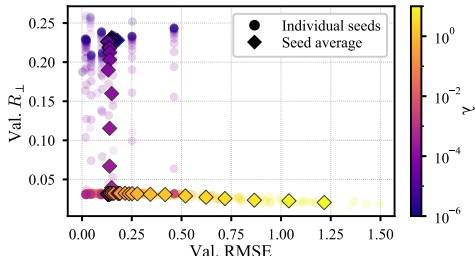

Figure 4: Trade-off curve for NeuralProphet model trained on step-function data.

Analogously to the above toy examples, we present a trade-off curve between RMSE and concurvity, averaging over 10 random initialization seeds per regularization strength $\lambda$. In this experiment, we choose $k = 400$ components for both daily and weekly seasonality, to allow concurvity to occur and fit the data almost exactly. Our findings are identical to the toy examples, demonstrating a steep decline in concurvity when increasing $\lambda$ with only a small increase in RMSE.

Finally, we note that concurvity can often be identified by visual inspection for additive univariate time-series models as each component is a function of the same variable, see Figure 1. In contrast, on multivariate tabular data, concurvity may go unnoticed if left unmeasured and hence lead to false conclusions, as we investigate next.

## 4.3 Tabular Data

In our final series of experiments, we investigate the benefits of the proposed regularizer when applied to NAMs trained on real-world tabular datasets – a domain often tackled with conventional machine learning methods such as random forests or gradient boosting. We concentrate our analysis on six well-studied datasets: Boston Housing [23], California Housing [37], Adult [18], MIMIC-II [29], MIMIC-III [27] and Support2 [14]. These datasets were selected with the aim of covering different dataset sizes (ranging between $N = 506$ for Boston Housing and $N = 48,842$ for Adult) as well as target variables (binary classification for Adult, MIMIC-II & -III, and Support2, and regression for California and Boston Housing). NAMs are used throughout the evaluation, subject to a distinct hyperparameter optimization for each dataset; more details are presented in Appendix B.1. Additionally, we provide a traditional, spline-based GAM using pyGAM [43] for comparison; details on the HPO for pyGAM can be found in Appendix B.2.

First, we explore the trade-off between the concurvity measure $R_\perp$ and validation fit quality when employing the proposed concurvity regularization. Figure 5 displays the results for the tabular datasets, including the pyGAM baseline. It is clear that the concurvity regularizer effectively reduces the concurvity measure $R_\perp$ without significantly compromising the model fit quality across all considered datasets, in particular in the case of small to moderate regularization strengths. For example, on the California Housing dataset, we are able to reduce $R_\perp$ by almost an order of magnitude from around 0.2 to 0.05, while the validation RMSE increases by about 10% from 0.59 to 0.66. Additionally, we

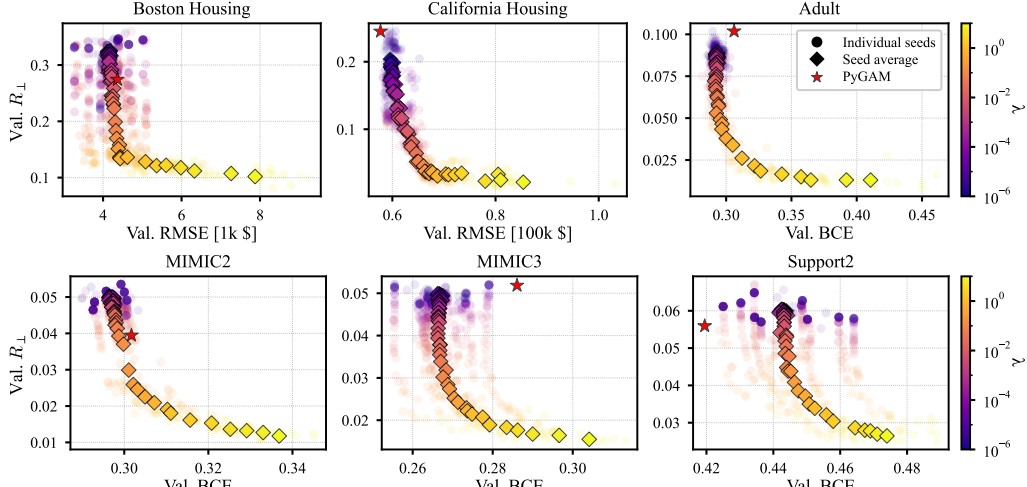

Figure 5: Trade-off curves between model fit quality and measured concurvity $R_\perp$ for 50 levels of concurvity regularization strength $\lambda$. Each regularization strength is evaluated over 10 initialization seeds to account for training variability, particularly noticeable in smaller datasets. The results of a conventional GAM are shown for comparison.

observe the variation in the scale of $R_\perp$ across the datasets, exemplified by the Adult dataset where the transformed features are nearly decorrelated even in the unregularized case, potentially implying a reduced necessity for regularization. In practice, trade-off curves between concurvity and model accuracy can serve as a valuable tool for identifying the optimal level of regularization strength. As expected, pyGAM obtains similar levels of concurvity as the unregularized NAM at equally similar levels of model performance. We refer to Appendix E.3.1 for more verbose trade-off curves.

**Case study: California Housing**    Our preceding experiments demonstrated that concurvity reduction can be achieved when training NAMs on tabular data. However, the practical significance of this observation in relation to interpretability remains unclear so far. To address this, we perform a more detailed analysis of NAMs trained on the California Housing dataset (see Appendix E.3.3 for similarly detailed results on the other datasets). In the following analysis, we compare NAMs trained with and without concurvity regularization. More specifically, we evaluate $\lambda = 0.1$ (determined based on Figure 5) and $\lambda = 0.0$ both for 60 random weight initializations.

First, we assess the effect of the regularizer on the model fit, finding that regularization increases the mean test RMSE by about 10% from about 0.58 to 0.64 and slightly decreases the spread between the seeds, as shown in Figure 6d. Note that the result in the non-regularized case is on par with the original NAM evaluation [2] serving as a sanity check of our experimental setup.

Second, we juxtapose the feature correlations of non-linearly transformed features for models trained with and without regularization. The results, as displayed in Figure 6a (upper right triangular matrices), are contrasted with the raw input feature correlations (lower left triangular matrices). It is evident that without regularization, high input correlations tend to result in correlated transformed features, as seen in the left correlation matrix. Conversely, the right correlation matrix reveals that concurvity regularization effectively reduces the correlation of transformed features. This effect is especially pronounced for previously highly correlated features such as *Longitude* and *Latitude*, or *Total Bedrooms* and *Households*.

Third, we investigate how concurvity impacts the estimation of the individual feature importances, which is of key interest for interpretable models such as NAMs. Following [2], we measure the importance of feature $i$ as $\frac{1}{N} \sum_{j=1}^{N} |f_i(x_{i,j}) - \overline{f_i}|$ where $\overline{f_i}$ denotes the average of the shape function $f_i$ over the training datapoints. We visualize the distribution of feature importances over our regularized and unregularized ensembles of NAMs in Figure 6b.[7] It is apparent that feature importances tend to have a larger variance in the unregularized case compared to the regularized case, a pattern which is

---

[7]We emphasize that the main purpose of Figure 6b is not a direct quantitative comparison of feature importances between different regularization levels but the discovery of qualitative insights.

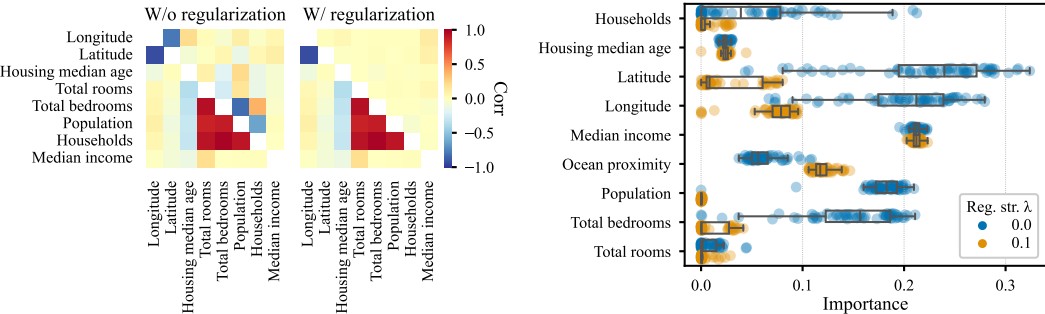

(a) Average feature correlations of the features (lower left) and non-linearly transformed features (upper right).

(b) Combined box and strip plot of the models' feature importances.

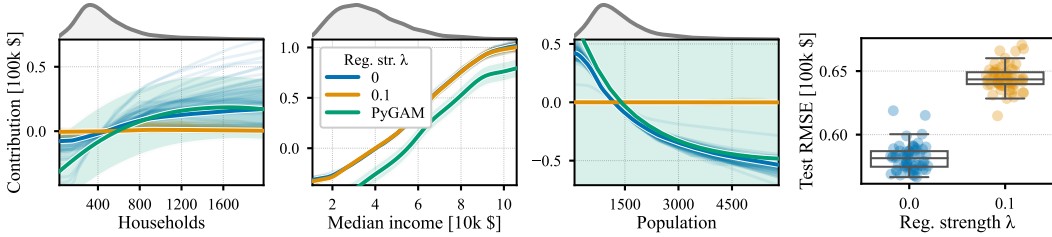

(c) Average and individual shape functions of 3 out of the 9 selected features. A kernel density estimate of the training data distribution is depicted on top. The results for pyGAM are shown for comparison, including 95% confidence intervals.

(d) Test RMSEs.

Figure 6: Results for the California Housing dataset. The considered NAMs were trained with and without concurvity regularization using 60 model initialization seeds each.

particularly clear for the strongly correlated features which we identified in Figure 6a. Such variance in feature importance can detrimentally impair the interpretability of the models, due to potential inconsistencies arising in absolute importance orders. However, our proposed concurvity regularizer effectively counteracts this issue, resulting in more consistent and compact feature importances across different random seeds. With regards to the varying effect of regularization on the respective features, two observations are particularly interesting: (1) Features that are mostly uncorrelated remain unaffected by the regularization – an effect we have previously seen in Toy Example 1 – which can, for example, be observed in the case of the *Median income* feature. (2) Input correlations lead to a bi-modal distribution in the corresponding feature importance as, for example, observable in the case of the *Longitude* and *Latitude* or *Total bedrooms* and *Households* features. Similarly to Toy Example 1, we see that the regularizer encourages feature selection for correlated feature pairs.

Finally, to visualize the impact of the regularization on model interpretability in more detail, the shape functions of three features are shown in Figure 6c. Here, the features *Households* and *Population* are strongly negatively correlated (see Figure 6a) which leads to their feature contributions largely canceling each other out. This problem is effectively mitigated by the proposed regularization, revealing naturally low contributions for both features. For comparison, the contribution of the mostly non-correlated *Median income* feature remains virtually unchanged by the regularization. Due to the concurvity in the data, the shape functions of the pyGAM model exhibit large variance which impedes their interpretability [40]. A similar behavior can be observed for the remaining feature contributions, which are depicted in Figure 11 in Appendix E.3.3.

In a supplementary experiment in Appendix E.3.4, we compare concurvity regularization to classical L1 regularization on the California Housing dataset. We find that L1 more strongly prunes features and removes finer details of each feature's contribution, which are more gently preserved for concurvity regularization.

In summary, our case study on the California Housing dataset establishes that concurvity regularization significantly enhances interpretability and consistency of a GAM in terms of shape functions and feature importance, whilst maintaining high model accuracy.

# 5   Related Work

**Classical works on concurvity in GAMs**    The term concurvity was first introduced in [9]; for a well-written introduction to concurvity, we refer the reader to [40]. Numerous subsequent works have developed techniques to address concurvity, such as improving numerical stability in spline-based GAM fitting [25, 48] and adapting Lasso regularization for GAMs [5]. Partial GAMs [21] were proposed to address concurvity through sequential maximization of Mutual Information between response variables and covariates. More recently, [28] compared several feature selection algorithms for GAMs and found algorithms selecting a larger feature set to be more susceptible to concurvity, a property first noticed in [22]. In addition, [28] proposes a novel feature selection algorithm that chooses a minimal subset to deal with concurvity. We refer to [28] for a thorough comparison of different concurvity measures. In contrast, our proposed regularizer adds no additional constraints on the feature set size and does not explicitly enforce feature sparsity.

**Modern neural approaches to GAMs**    Recent advancements in neural approaches to GAMs, such as Neural Additive Models (NAMs) [2] and NeuralProphet [46], have provided more flexible and powerful alternatives to classical methods [24]. These have spurred interest in the subject leading to several extensions of NAMs [11, 19, 39]. Our approach is compatible with the existing methodologies and can be readily integrated if they are implemented in an automatic differentiation framework.

**Regularization via decorrelation**    Similar types of decorrelation regularizers have previously been proposed in the machine learning literature but in different contexts. [13] found that regularizing the cross-covariance of hidden activations significantly increases generalization performance and proposed DeCov. OrthoReg [41] was proposed to regularize negatively correlated features to increase generalization performance by reducing redundancy in the network. Similarly, [49] proposes to add a regularizer enforcing orthonormal columns in weight matrices. More recent approaches, such as Barlow Twins [51], leverage decorrelation as a self-supervised learning technique to learn representations that are invariant to different transformations of the input data.

# 6   Conclusion

In this paper, we have introduced a differentiable concurvity regularizer, designed to mitigate the often overlooked issue of concurvity in differentiable Generalized Additive Models (GAMs). Through comprehensive empirical evaluations, we demonstrated that our regularizer effectively reduces concurvity in differentiable GAMs such as Neural Additive Models and NeuralProphet. This in turn significantly enhances the interpretability and reliability of the learned feature functions, a vital attribute in various safety-critical and strongly regulated applications. Importantly, our regularizer achieves these improvements while maintaining high prediction quality, provided it is carefully applied. We underscore that while the interpretability-accuracy trade-off is an inherent aspect of concurvity regularization, the benefits of increased interpretability and consistent feature importances across model initializations are substantial, particularly in real-world decision-making scenarios.

Nonetheless, our study is not without its limitations. The validation of our approach, while diverse, was limited to three real-world and three synthetic datasets. As such, we acknowledge that our findings may not fully generalize across all possible datasets and use cases.

An intriguing avenue for future work could be to examine the impact of our regularizer on fairness in GAMs. While prior work [12] suggests that GAMs with high feature sparsity can miss patterns in the data and be unfair to minorities, our concurvity regularizer does not directly enforce feature sparsity. Thus, a comparison between sparsity regularizers and our concurvity regularizer in unbalanced datasets would be of high interest. In addition, future work could explore how the joint optimization of concurvity and model fit could be improved by framing it as a multi-objective problem. Finally, our concurvity regularizer could be adapted to differentiable GAMs that incorporate pairwise interactions. Specifically, it would be interesting to contrast such an extension with the ANOVA decomposition proposed in [30] in terms of single and pairwise interactions.

We conclude by encouraging researchers and practitioners to "curve your enthusiasm" – that is, to seriously consider concurvity in GAM modeling workflows. We believe this will lead to more interpretable models and hence more reliable and robust analyses, potentially avoiding false conclusions.

## Acknowledgements

This work has received funding from the German Federal Ministry for Economic Affairs and Climate Action as part of the ResKriVer project under grant no. 01MK21006H. We would like to thank Roland Zimmermann, Fabio Haenel, Aaron Klein, Christian Leibig, Felix Pieper, Sebastian Schulze, Ziyad Sheebaelhamd, and Thomas Wollmann for their constructive feedback in the preparation of this paper.

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

# A  Additional Remarks and Theoretical Results for the Proposed Regularizer

## A.1  The Decorrelation Space $\mathcal{H}_\perp$ Rules Out Concurvity

In this section, we formalize our claim from Section 3 that the space $\mathcal{H}_\perp$ provably rules out concurvity. In the following, $\langle \cdot, \cdot \rangle$ and $\| \cdot \|_2$ denote the Euclidean scalar product and norm, respectively. For $v = (v_1, \ldots, v_N) \in \mathbb{R}^N$, we set $\bar{v} := \frac{1}{N} \sum_{l=1}^{N} v_l$ and denote the all-one vector by $\mathbb{1} := (1, \ldots, 1) \in \mathbb{R}^N$. The standard (empirical) Pearson's correlation coefficient is then given by[8]

$$\mathrm{Corr}(v, w) := \begin{cases} \frac{\langle v - \bar{v}\mathbb{1}, w - \bar{w}\mathbb{1} \rangle}{\|v - \bar{v}\mathbb{1}\|_2 \cdot \|w - \bar{w}\mathbb{1}\|_2} & \text{if } v \text{ and } w \text{ are non-constant,} \\ \infty & \text{otherwise,} \end{cases} \quad v, w \in \mathbb{R}^N.$$

**Lemma A.1.** *Let* $X_1, \ldots, X_p \in \mathbb{R}^N$ *be a set of feature variables with* $p > 1$ *and let* $\mathcal{H} \subset \{(f_1, \ldots, f_p) \mid f_i : \mathbb{R} \to \mathbb{R}\}$ *be a class of functions. Consider the following subclass of* $\mathcal{H}$:

$$\mathcal{H}_\perp := \{(f_1, \ldots, f_p) \in \mathcal{H} \mid \mathrm{Corr}(f_i(X_i), f_j(X_j)) = 0 \text{ for all } i \neq j \} \subset \mathcal{H}.$$

*Then we do* not *have concurvity w.r.t.* $X_1, \ldots, X_p$ *and* $\mathcal{H}_\perp$:

*Proof.* Towards a contradiction, assume that there are $(g_1, \ldots, g_p) \in \mathcal{H}_\perp$ and $c_0 \in \mathbb{R}$ such that $c_0 \mathbb{1} + \sum_{i=1}^{p} g_i(X_i) = 0$.

We set $v_i := g_i(X_i)$ and trivially add the mean vectors to the linear combination:

$$0 = c_0 \mathbb{1} + \sum_{i=1}^{p} v_i = \left( c_0 + \sum_{i=1}^{p} \bar{v}_i \right) \mathbb{1} + \sum_{i=1}^{p} (v_i - \bar{v}_i \mathbb{1}).$$

Using that $\mathrm{Corr}(v_i, v_j) = 0$ for $i \neq j$, we then obtain

$$0 = \langle 0, v_j - \bar{v}_j \mathbb{1} \rangle = \left( c_0 + \sum_{i=1}^{p} \bar{v}_i \right) \langle \mathbb{1}, v_j - \bar{v}_j \mathbb{1} \rangle + \sum_{i=1}^{p} \langle v_i - \bar{v}_i \mathbb{1}, v_j - \bar{v}_j \mathbb{1} \rangle$$

$$= \left( c_0 + \sum_{i=1}^{p} \bar{v}_i \right) \cdot (N\bar{v}_j - N\bar{v}_j) + \|v_j - \bar{v}_j \mathbb{1}\|_2^2 = \|v_j - \bar{v}_j \mathbb{1}\|_2^2.$$

We conclude that $v_j = \bar{v}_j \mathbb{1}$, i.e., $v_j$ is a constant vector. But this contradicts the definition of $\mathcal{H}_\perp$ because we would have $\mathrm{Corr}(v_i, v_j) = \infty$ with any $i \neq j$. $\square$

## A.2  Additional Remarks on Concurvity and our Regularizer

(1) The definitions of multicollinearity (Definition 2.1) and concurvity (Definition 2.2) are based on a fixed (deterministic) feature design, but one could also formulate probabilistic versions, cf. [44]. The latter typically facilitates a theoretical analysis, which, however, is not the focus of our work. Moreover, a probabilistic definition would not cover an important practical source of multicollinearity, namely underdetermined systems where $N < p$.

(2) Although closely related, multicollinearity does not necessarily imply concurvity and vice versa. Indeed, one can easily come up with setups where perfectly correlated features become decorrelated after a non-linear transform. Similarly, uncorrelated input features can be made perfectly correlated with a non-linearity. Two simple (toy) examples are presented in Section 4.1.

(3) Our concurvity regularizer $R_\perp$ does not automatically affect the predictive performance of a GAM. For example, assuming that the input features are drawn from stochastically independent random variables, we can conclude that $\mathrm{Corr}(f_i(X_i), f_j(X_j)) \approx 0$ for a large enough sample size $N$, since non-linear transforms of independent random variables remain independent. Consequently, we have that $R_\perp(\{f_i\}_i, \{X_i\}_i) \approx 0$, so that no (in this case undesirable) regularization takes effect.

---

[8]The special case of constant vectors could be treated differently, e.g., by setting the correlation to 0. The version we use here is most convenient for our purposes as it excludes the treatment of additional special cases in Lemma A.1.

# B Hyperparameter Optimization

## B.1 Tabular Datasets

For hyperparameter optimization we use Tree-structured Parzen Estimator (TPE) [7] as implemented in Optuna [3]. We run the optimization for a budget of 500 function evaluations and optimize w.r.t. validation RMSE for Boston Housing and California Housing or validation binary cross entropy loss for Adult, MIMIC-2, MIMIC-3, and Support 2.

The hyperparameter space and default parameters are shown in Table 1 and the hyperparameters per dataset are shown in Figure 7.

| Hyperparameter | Value / Range | Scaling |
|---|---|---|
| Learning Rate | [1e-4, 1e-1] | log |
| Weight Decay | [1e-6, 1] | log |
| Activation | [ELU, GELU, ReLU] | cat. |
| # of neurons per layer | [2, 256] | linear |
| # of hidden layers | [1, 6] | linear |
| Num. Epochs | [10, 500] | linear |

Table 1: Hyperparameter Search Space

| Hyperparameter | Value |
|---|---|
| Learning Rate | 1e-3 |
| Weight Decay | 0.0 |
| Activation | GELU |
| # of neurons per layer | 128 |
| # of hidden layers | 3 |
| Num. Epochs | 50 |
| Batch Size | 128 |
| Correlation Denominator $\epsilon$ | 1e-12 |
| Start Conc. Reg. after x% of steps | 5 |

(a) Toy Example 1&2

| Hyperparameter | Value |
|---|---|
| Learning Rate | 7.93e-4 |
| Weight Decay | 1.79e-2 |
| Activation | ELU |
| # of neurons per layer | 75 |
| # of hidden layers | 6 |
| Num. Epochs | 91 |
| Batch Size | 128 |
| Correlation Denominator $\epsilon$ | 1e-12 |
| Start Conc. Reg. after x% of steps | 5 |

(b) Boston Housing

| Hyperparameter | Value |
|---|---|
| Learning Rate | 9.46e-3 |
| Weight Decay | 3.73e-3 |
| Activation | ReLU |
| # of neurons per layer | 72 |
| # of hidden layers | 5 |
| Num. Epochs | 39 |
| Batch Size | 512 |
| Correlation Denominator $\epsilon$ | 1e-12 |
| Start Conc. Reg. after x% of steps | 5 |

(c) California Housing

| Hyperparameter | Value |
|---|---|
| Learning Rate | 2.64e-3 |
| Weight Decay | 1.64e-3 |
| Activation | GELU |
| # of neurons per layer | 204 |
| # of hidden layers | 4 |
| Num. Epochs | 200 |
| Batch Size | 512 |
| Correlation Denominator $\epsilon$ | 1e-12 |
| Start Conc. Reg. after x% of steps | 5 |

(d) Adult

| Hyperparameter | Value |
|---|---|
| Learning Rate | 3.31e-3 |
| Weight Decay | 1.08e-3 |
| Activation | GELU |
| # of neurons per layer | 190 |
| # of hidden layers | 3 |
| Num. Epochs | 20 |
| Batch Size | 512 |
| Correlation Denominator $\epsilon$ | 1e-12 |
| Start Conc. Reg. after x% of steps | 5 |

(e) MIMIC-II

| Hyperparameter | Value |
|---|---|
| Learning Rate | 3.88e-3 |
| Weight Decay | 1.10e-3 |
| Activation | GELU |
| # of neurons per layer | 168 |
| # of hidden layers | 3 |
| Num. Epochs | 23 |
| Batch Size | 512 |
| Correlation Denominator $\epsilon$ | 1e-12 |
| Start Conc. Reg. after x% of steps | 5 |

(f) MIMIC-III

| Hyperparameter | Value |
|---|---|
| Learning Rate | 3.42e-3 |
| Weight Decay | 1.05e-2 |
| Activation | GELU |
| # of neurons per layer | 127 |
| # of hidden layers | 3 |
| Num. Epochs | 30 |
| Batch Size | 512 |
| Correlation Denominator $\epsilon$ | 1e-12 |
| Start Conc. Reg. after x% of steps | 5 |

(g) Support2

Figure 7: Hyperparameters per dataset for NAM experiments.

## B.2 pyGAM

We optimized the second-order derivative penalties on the spline functions in pyGAM [43] using random search. We used a budget of 300 function evaluations within a search range of [1e-3, 1e3].

# C Experimental Details

## C.1 NAM Experiments

In all of our NAM experiments, we use the AdamW optimizer [33] and adjust our learning rate using Cosine Annealing [32] and decay to 0. For all regression problems we use the Mean Squared Error (MSE) and for all binary classfication problems the Binary Cross Entropy (BCE) as loss function $L$.

In all of our experiments, we apply the concurvity regularization after a warm-up phase consisting of 5% of the total optimization steps for stability reasons. After this warm-up phase, regularization is performed at each step.

## C.2 Toy Examples

We sample 10,000 datapoints from the model and use 7,000, 2,000, 1,000 for training, validation and testing respectively. We use the validation split to find adequate hyperparameters via a small manual search. Our NAM has 3 layers per feature NN with 128 hidden units and uses the GeLU [26] activation function. We use no weight decay, train for 50 epochs at an initial learning rate of 1e-3 and a batch size of 128. Otherwise, the experimental setup is the same as described earlier in Section C.1.

# D   Dataset Details

**Boston Housing**   The Boston Housing Dataset [23], compiled by Harrison and Rubinfeld in the 1970s, is a benchmark dataset employed in machine learning and statistical modeling for housing price prediction. Consisting of 506 neighborhoods in the Boston metropolitan area, it features 13 attributes, including crime rate, zoning information, industrial acreage, Charles River proximity, air quality, housing characteristics, accessibility to employment centers, highways, and education, as well as demographic factors. The primary objective is to predict the median value of owner-occupied homes using these features.

**California Housing**   The California Housing Dataset [37] is a widely-used benchmark dataset for machine learning and statistical modeling, particularly in the domain of housing price prediction. Originally derived from the 1990 California Census, it consists of 20,640 samples, each representing a census block group. The dataset contains information on various housing-related attributes, such as median income, housing median age, average number of rooms, average number of bedrooms, population and average household size. It also includes the geographical location (latitude and longitude) of each block group. The objective is to predict the median house value. We obtained the dataset from [18].

**Adult**   The Adult Dataset [18] is also known as the "Census Income" dataset. Extracted from the 1994 United States Census Bureau data, it comprises 48,842 records, each representing an individual. The dataset contains 14 features, including age, work class, education, marital status, occupation, relationship, race, sex, capital gain, capital loss, hours worked per week, and native country. The objective is to predict whether an individual's annual income exceeds $50,000.

**MIMIC-II**   The MIMIC-II (Multiparameter Intelligent Monitoring in Intensive Care) dataset [29] is a public database, offering clinical data from a multitude of Intensive Care Unit (ICU) patients. It is managed by the MIT Lab for Computational Physiology and encompasses a wide variety of data points, such as patient demographics, vital signs, lab test results, medications, procedures, caregiver notes, and imaging reports, as well as mortality rates both in and out of the hospital.

**MIMIC-III**   MIMIC-III (Medical Information Mart for Intensive Care III) [27] is the successor to the MIMIC-II database. It contains additional, more recent patient records and provides more detailed data, including free-text interpretation of imaging reports, allowing for more granular research and improved application in areas like machine learning, health informatics, and predictive modeling.

**SUPPORT2**   The Support2 [14] (Study to Understand Prognoses and Preferences for Outcomes and Risks of Treatments) dataset is a clinical database that contains detailed medical information from a large cohort of seriously ill hospitalized adults. The dataset was created as part of a multi-center study designed to understand the outcomes of decisions made in the course of medical treatment. It provides extensive variables, including demographic data, physiological measurements, diagnostic information, treatment plans, and outcomes such as survival and quality of life. The data collected span a diverse set of medical conditions, making the SUPPORT2 dataset a valuable resource for researchers seeking to study clinical decision-making, prognosis evaluation, and healthcare outcomes.

# E  Additional results

## E.1  Toy Example 1

Figure 8 shows an additional scatter plot comparing the contributions of the transformed features.

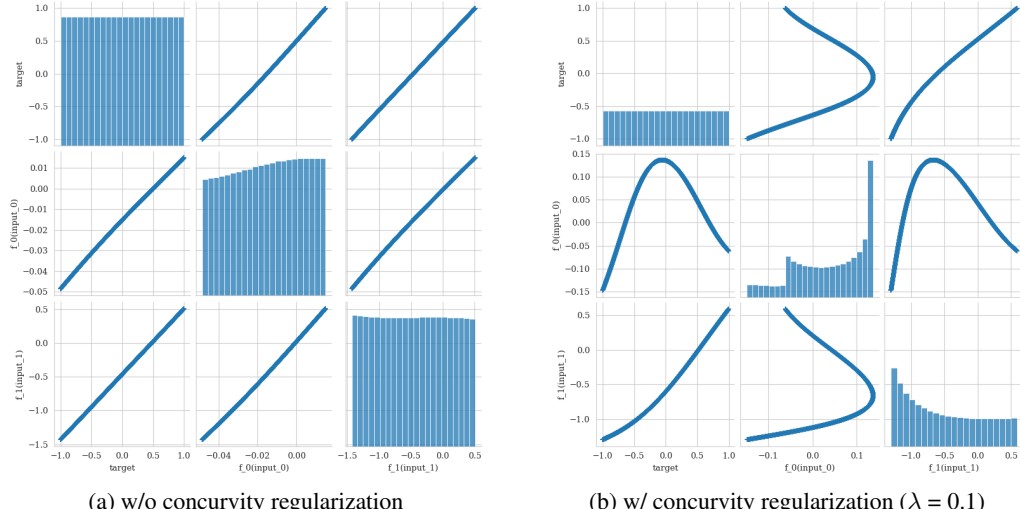

(a) w/o concurvity regularization

(b) w/ concurvity regularization ($\lambda = 0.1$)

Figure 8: (Toy Example 1) Pair plot demonstrating the difference in shape functions learned without and with concurvity regularization. The features $X_1$ and $X_2$ are fully correlated in this example. Each subplot is generated by a scatter plot over a large set of random samples from the input feature pair $(X_1, X_2)$. As expected, the model trained without regularization shows strong (almost perfect) linear correlation between the transformed features $f_1(X_1)$, $f_2(X_2)$ and the target variable $Y = X_1$, while our regularizer effectively decorrelates $f_1(X_1)$ and $f_2(X_2)$. But note that $f_1(X_1)$ and $f_2(X_2)$ are not stochastically independent, as the plots in (b) clearly show a non-linear functional relationship between the two features.

### E.2 Additional Toy Example from [28]

In response to a suggestion to include a more complex toy example during the review process, we have replicated the toy example from [28] using our NAM setup. To recap, this example contains 7 input features:

$$
\begin{aligned}
X_1 &\sim X_2 \sim X_3 \sim U(0,1), \\
X_4 &= X_2^3 + X_3^2 + N(0,\sigma_1), \\
X_5 &= X_3^2 + N(0,\sigma_1), \\
X_6 &= X_2^2 + X_4^2 + N(0,\sigma_1), \\
X_7 &= X_1 \cdot X_2 + N(0,\sigma_1), \\
Y &= 2X_1^2 + X_5^3 + 2\sin(X_6) + N(0,\sigma_2),
\end{aligned}
$$

where the standard deviations are sufficiently small to create severe concurvity among the features ($\sigma_1 = 0.05, \sigma_2 = 0.5$). We simulated $10,000$ data points from this model and created a 7:3 train/test split. We fitted 20 random initializations of a NAM in the unregularized, concurvity-regularized, and L1 regularized setting. The regularization parameter was determined separately for each regularization type based on trade-off curves. For concurvity regularization, we used $\lambda = 0.1$, and for L1, we used $\lambda = 0.05$.

The results, reported as the $R^2$ on the test set, are reported in Table 2, with the top three rows from [28] for comparison. Confidence intervals of the mean are estimated on $10,000$ bootstrap samples. Features are presented in descending order of their importance (as defined in the main paper) for the best-fitting model in each setting, while the actual importances are reported below.

| Model | Selected features | $R^2$ (%) (5%, 95%) | $R_\perp$ (5%, 95%) |
|---|---|---|---|
| Full model [28] | Full model | 84.99 | |
| Stepwise [28] | $\mathbf{X_1}, X_4, \mathbf{X_5}, \mathbf{X_6}$ | 85.11 | |
| Hybrid [28] | $\mathbf{X_1}, \mathbf{X_5}, \mathbf{X_6}$ | 85.31 | |
| Unregularized (ours) $\rightarrow$ Feature importance | $\mathbf{X_1}, \mathbf{X_6}, X_4, X_2, \mathbf{X_5}, X_3, X_7$ $\mathbf{0.129}, \mathbf{0.097}, 0.066, 0.053, \mathbf{0.043}, 0.013, 0.004$ | 80.77 (80.31 / 80.95) | 0.22 (0.20 / 0.23) |
| Concurvity Reg. (ours) $\rightarrow$ Feature importance | $\mathbf{X_1}, \mathbf{X_6}, \mathbf{X_5}, X_2, X_7, X_4, X_3$ $\mathbf{0.132}, \mathbf{0.125}, \mathbf{0.088}, 0.070, 0.006, 0.002, 0.002$ | 79.28 (78.52 / 79.88) | 0.03 (0.02 / 0.03) |
| L1 Reg. (ours) $\rightarrow$ Feature importance | $\mathbf{X_6}, \mathbf{X_1}, \mathbf{X_5}, X_7, X_4, X_3, X_2$ $\mathbf{0.147}, \mathbf{0.106}, \mathbf{0.037}, 0.009, 0.008, 0.005, 0.0$ | 79.12, (78.50 / 79.47) | 0.21, (0.20 / 0.21) |

Table 2: Results of replication experiment of the toy example proposed in [28]. All results are calculated over the test split, with our results reported as mean values and confidence intervals. In addition to our three NAM settings (unregularized, concurvity regularization, and L1 regularization), we report the values from the original paper [28] for comparison.

We note that both concurvity regularization and L1 regularization correctly identify the three predictive features $X_1, X_5$ and $X_6$, on which $Y$ directly depends. This is not the case without regularization. Furthermore, we find that concurvity regularization effectively reduces $R_\perp$, unlike L1 regularization.

We also note that the $R^2$ values of all NAM implementations are lower than those reported in [28], which we believe is due to the inductive biases in spline-based models being particularly well-suited to the mostly polynomial-based problem. This notwithstanding, our results further underline the effectiveness of concurvity regularization.

### E.3   Tabular Data

#### E.3.1   Verbose Trade-off Curves

In addition to the trade-off curves presented in the main text (see Figure 5), Figure 9 shows a more verbose version, separating the visualization for the measured validation concurvity and validation accuracy as a function of the regularization strength $\lambda$. This explicit visualization allows us for an easier choice of $\lambda$, for example, by determining the elbow in the validation accuracies, after which stronger regularization would lead to a significant loss of performance. For example, this can be nicely observed in the case of the Boston Housing dataset at $\lambda \approx 1$.

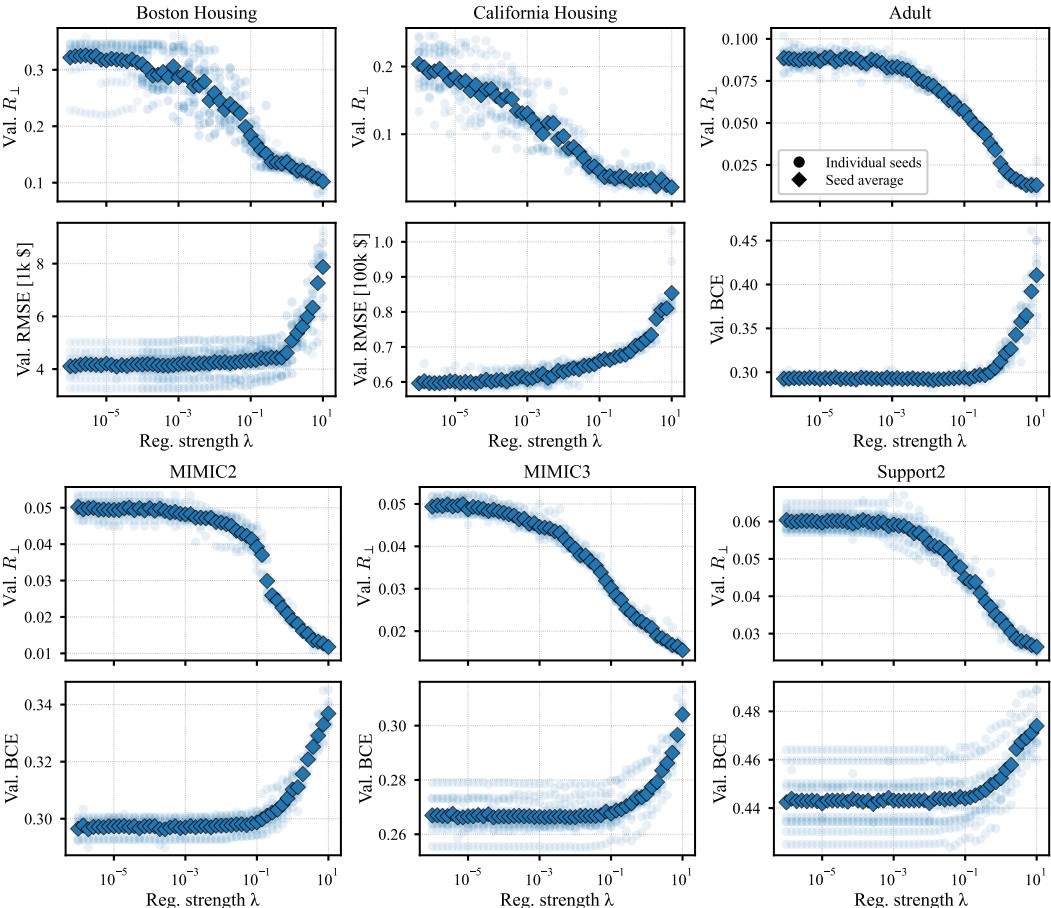

Figure 9: Verbose trade-off curves between the model fit quality and measured concurvity $R_\perp$ for 50 levels of the concurvity regularization strength $\lambda$. Each regularization strength is evaluated over 10 initialization seeds to account for training variability, which is particularly noticeable in smaller datasets.

#### E.3.2   Runtime Analysis

Due to calculating pair-wise correlations, our proposed regularizer has a quadratic computational complexity. However, using an efficient implementation leveraging vectorization, the scaling constant can be well controlled. To investigate the computational overhead of our regularizer, we measure the mean absolute runtime across 100 repetitions for different numbers of input features and batch sizes. Furthermore, we evaluate the relative overhead of our regularizer compared to a NAM model. The relative overhead clearly depends on the size and implementation details of the NAM. Here, we compare with a NAM for which each feature-dependent network consists of a 3-layer MLP with 128 neurons each. The calculation over the features is vectorized using batched `matmuls`. The results are shown in Figure 10, all obtained with an M1 MacBook Pro. As expected, we observe a power-law

scaling of the runtime of the regularizer, which is in particular observable for an increasing numbers of input features. In particular, for moderate numbers of input features between 64 and 256, the absolute runtime is about 1 ms. Compared with the forward pass of a 3-layer NAM, the proposed regularizer typically implies a relative overhead of well below 10 %. Note that the results compare only with the forward pass of the NAM model and do not include data loading, batch collation, loss calculation, and backward passes. We note that especially for larger batch sizes, the relative overhead introduced by our regularizer is well below 5 %. On the other hand, for smaller batch sizes, the relative overhead can become large, due to the reduced effectiveness of vectorization. In absolute terms, however, the overhead is very small for smaller batch sizes as well. We therefore conclude that the overhead introduced by regularization is negligible in commonly considered model setups.

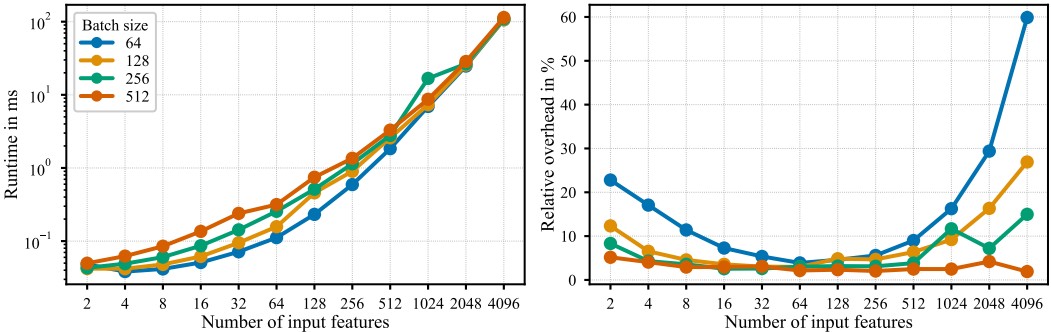

Figure 10: Left plot: Absolute runtime of the proposed regularizer for different numbers of input features and batch sizes. Right plot: Computational overhead relative to the forward pass of a 3-layer NAM, with 128 neurons each, for different numbers of input features and batch sizes.

### E.3.3   In-detail Results for the Remaining Tabular Datasets

In addition to the results of our case study with the California Housing dataset in Section 4.3, we here present the shape functions of the complete set of features, see Figure 11.

Furthermore, we show the analogous results for the remaining datasets Adult (Figure 12), Boston Housing (Figure 13), MIMIC-II (Figure 14), MIMIC-III (Figure 15 & 16), and Support2 (Figure 17 & 18).

### E.3.4   Comparing Concurvity Regularization to L1 Regularization

Although we chose pyGAM as our baseline in the main paper, we have additionally compared our concurvity regularizer to L1 regularization, as a popular alternative to eliminating redundant features. Results on the California Housing dataset are included in Figure 19. We performed L1 regularization on the terms $w_i f_i(x_i)$ in Equation (GAM), with regularization strength $\lambda = 0.1$, as determined by trade-off curves on feature L1 norms versus RMSE.

The figure suggests that L1 regularization prunes features more strongly than concurvity regularization, which may be desirable in some situations. However, this comes at a cost of a larger test RMSE, as well as some seemingly predictive features being pruned. For example, L1 regularization removes the two local peaks in the Latitude dimension corresponding to San Francisco and Los Angeles. In addition, unlike L1 regularization, concurvity regularization does not prune uncorrelated features.

Finally, we note that there may be different motivations behind choosing sparsity regularization (as few features as possible) or concurvity regularization (features become as decorrelated as possible). If the main goal is to address concurvity, then sparsity regularization is a blunt tool, and vice versa.

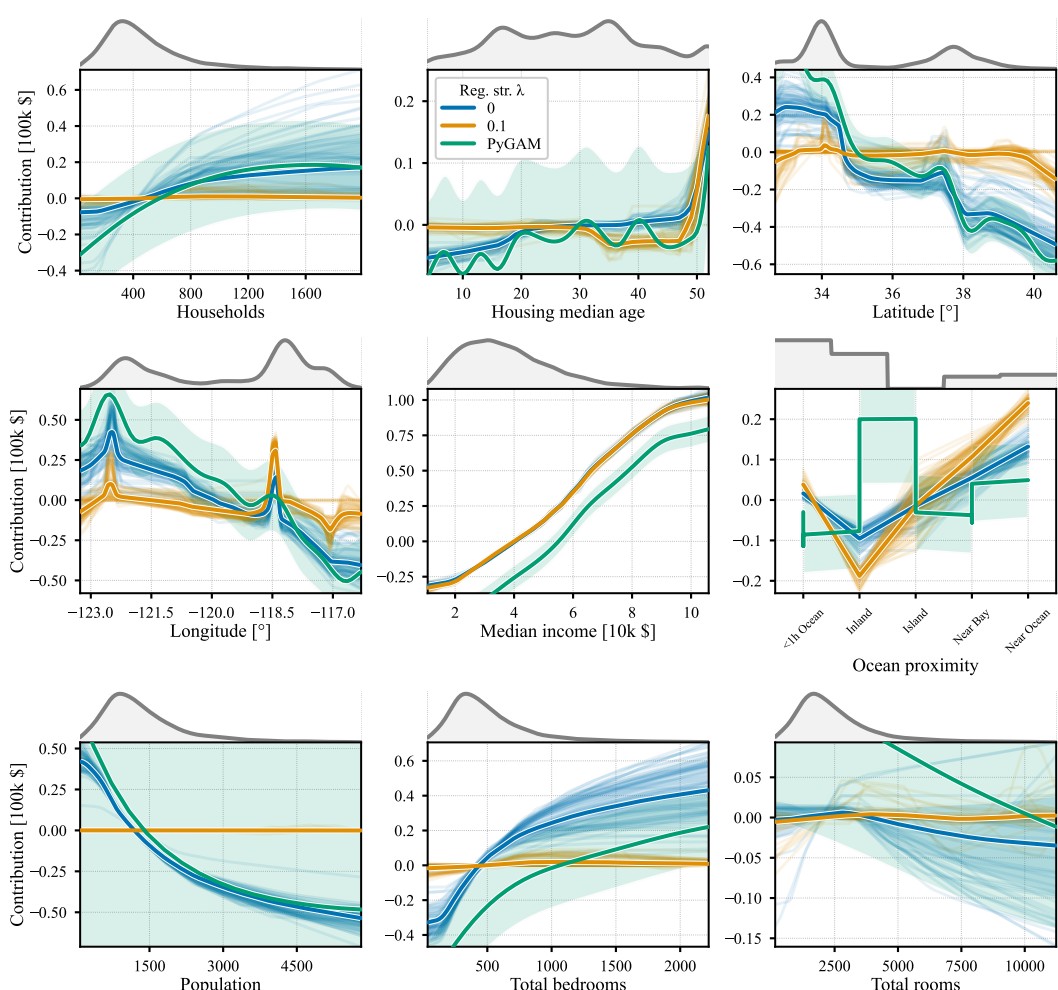

Figure 11: Shape functions of all input features in the case of the California Housing dataset. Depicted are the average as well as individual contributions with and without concurvity regularization using 60 initialization seeds each. The pyGAM results plotted for comparison include the estimated 95% confidence intervals.

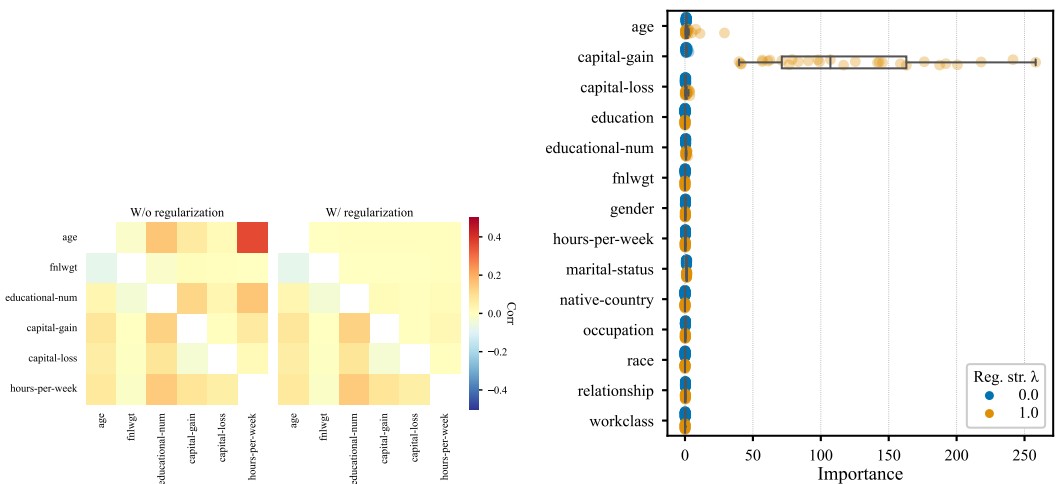

(a) Average feature correlations of the features (lower left) and non-linearly transformed features (upper right). Showing non-categorical features only.

(b) Combined box and strip plot of the models' feature importances.

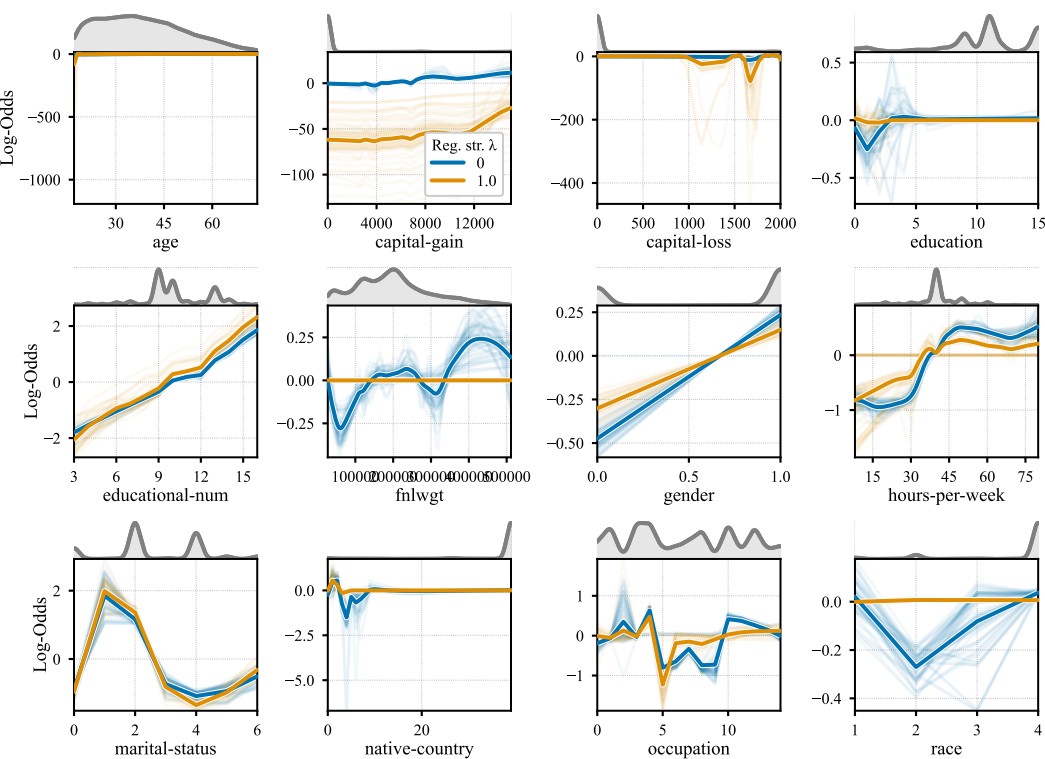

(c) Average and individual shape functions. A kernel density estimate of the training data distribution is depicted on top.

Figure 12: Results for the **Adult dataset**. The considered NAMs were trained with and without concurvity regularization using 60 model initialization seeds each.

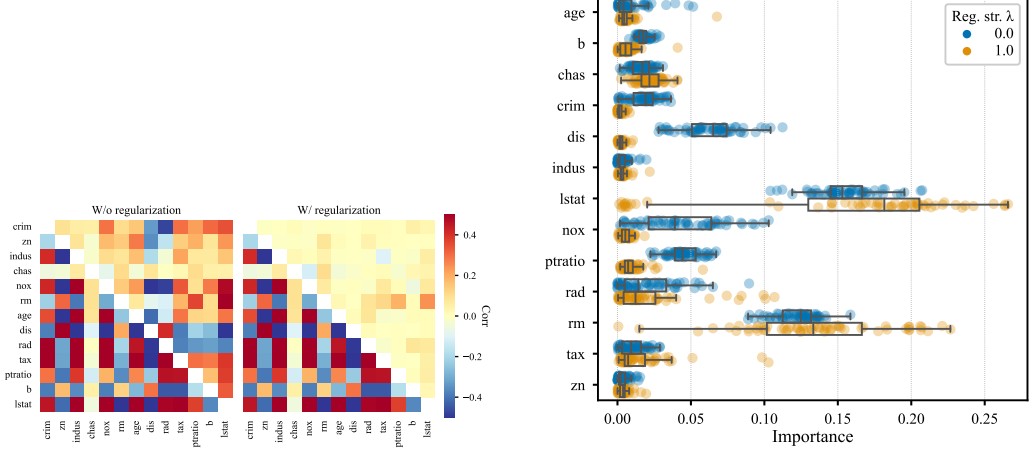

(a) Average feature correlations of the features (lower left) and non-linearly transformed features (upper right).

(b) Combined box and strip plot of the models' feature importances.

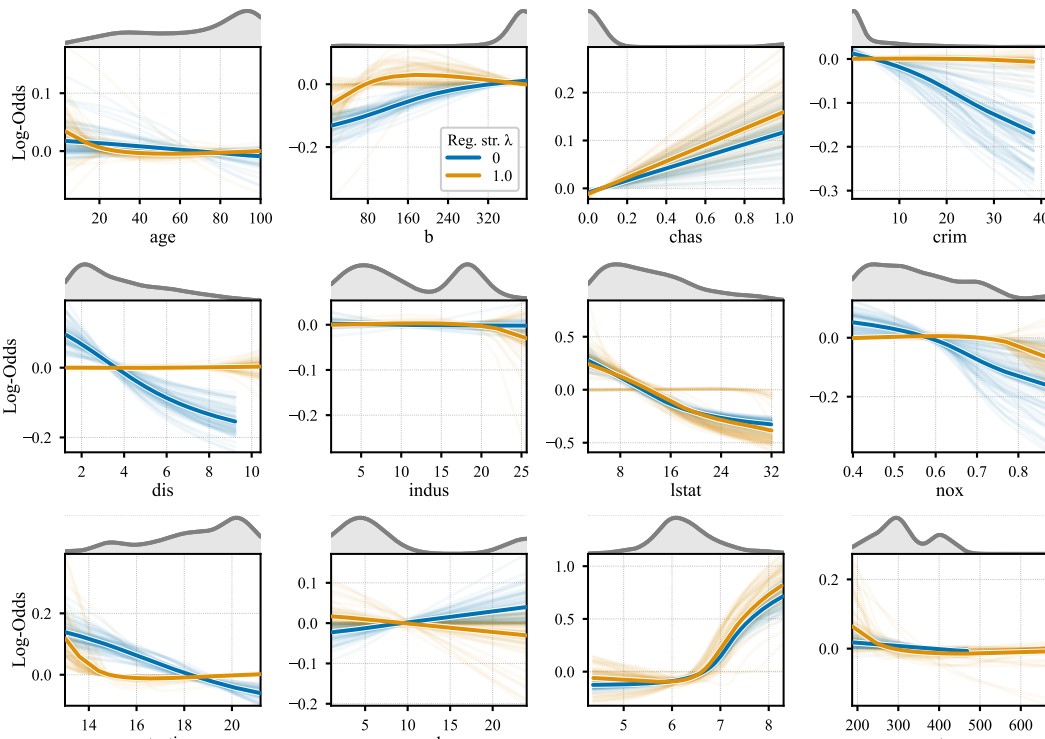

(c) Average and individual shape functions. A kernel density estimate of the training data distribution is depicted on top.

Figure 13: Results for the **Boston Housing dataset**. The considered NAMs were trained with and without concurvity regularization using 60 model initialization seeds each.

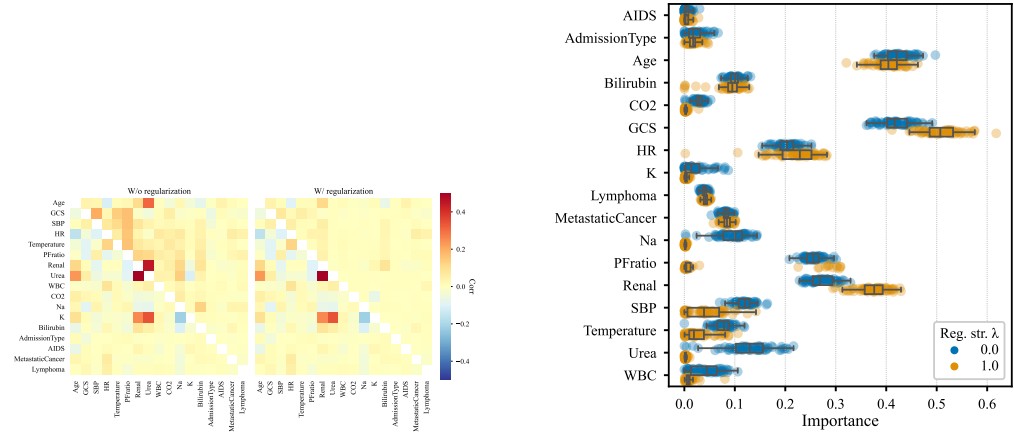

(a) Average feature correlations of the features (lower left) and non-linearly transformed features (upper right).

(b) Combined box and strip plot of the models' feature importances.

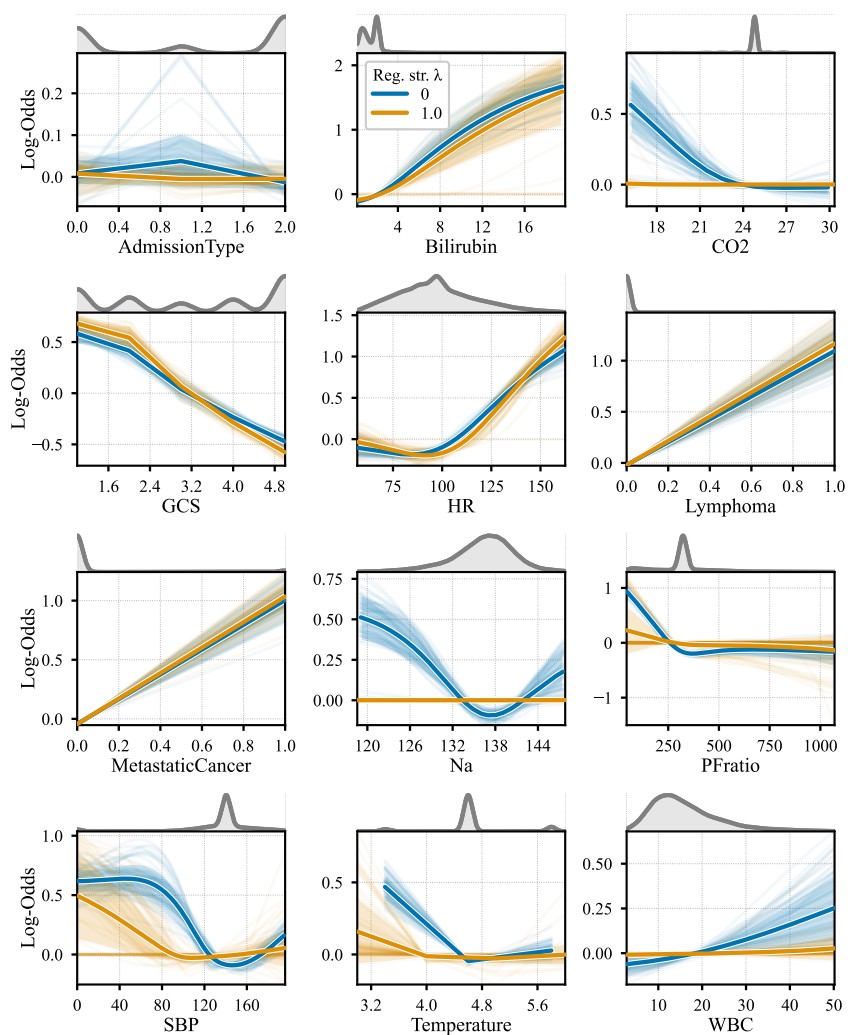

(c) Average and individual shape functions. A kernel density estimate of the training data distribution is depicted on top.

Figure 14: Results for the **MIMIC-II dataset**. The considered NAMs were trained with and without concurvity regularization using 60 model initialization seeds each.

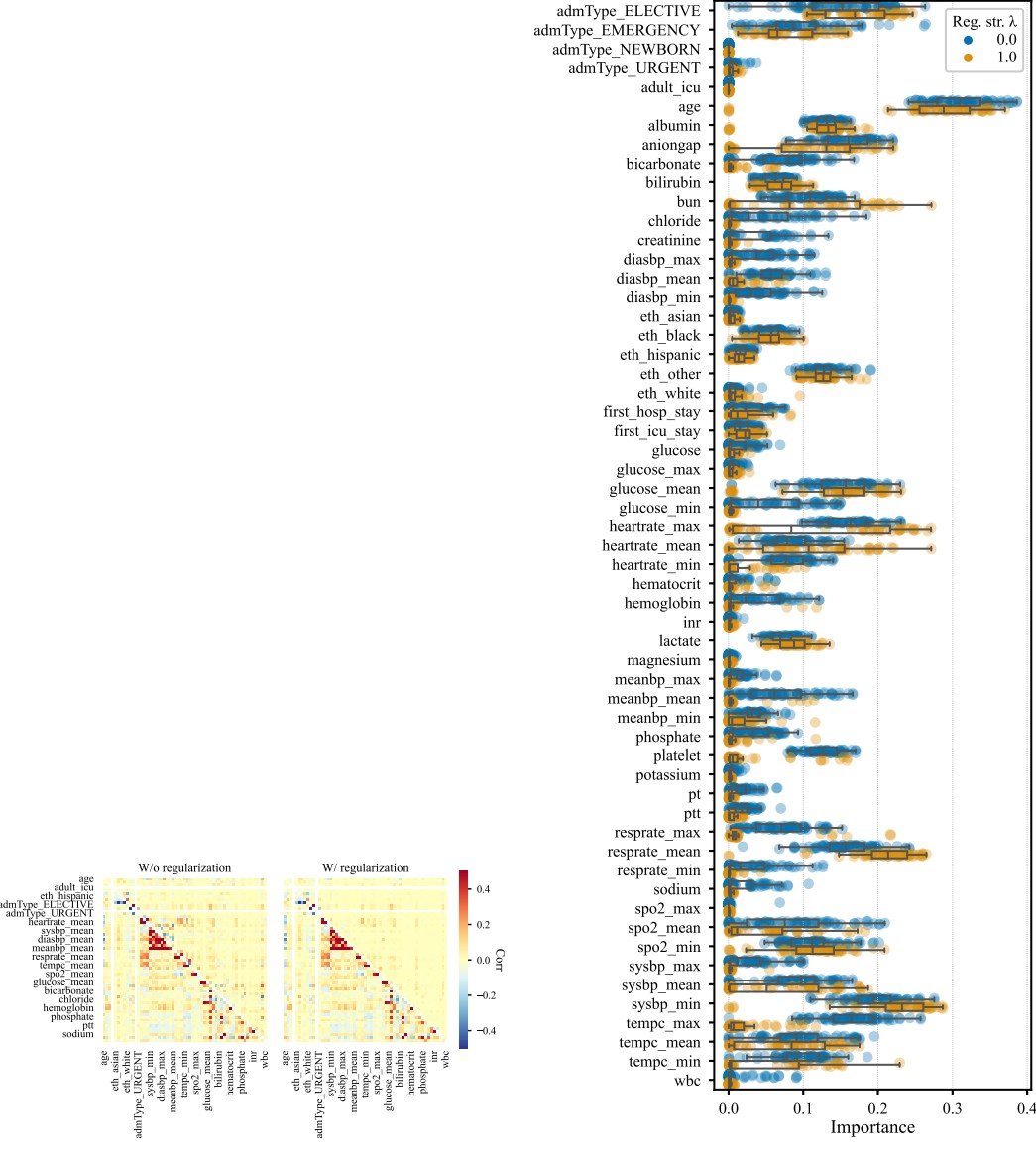

(a) Average feature correlations of the features (lower left) and non-linearly transformed features (upper right).

(b) Combined box and strip plot of the models' feature importances.

Figure 15: Results (part 1) for the **MIMIC-III dataset**. The considered NAMs were trained with and without concurvity regularization using 60 model initialization seeds each.

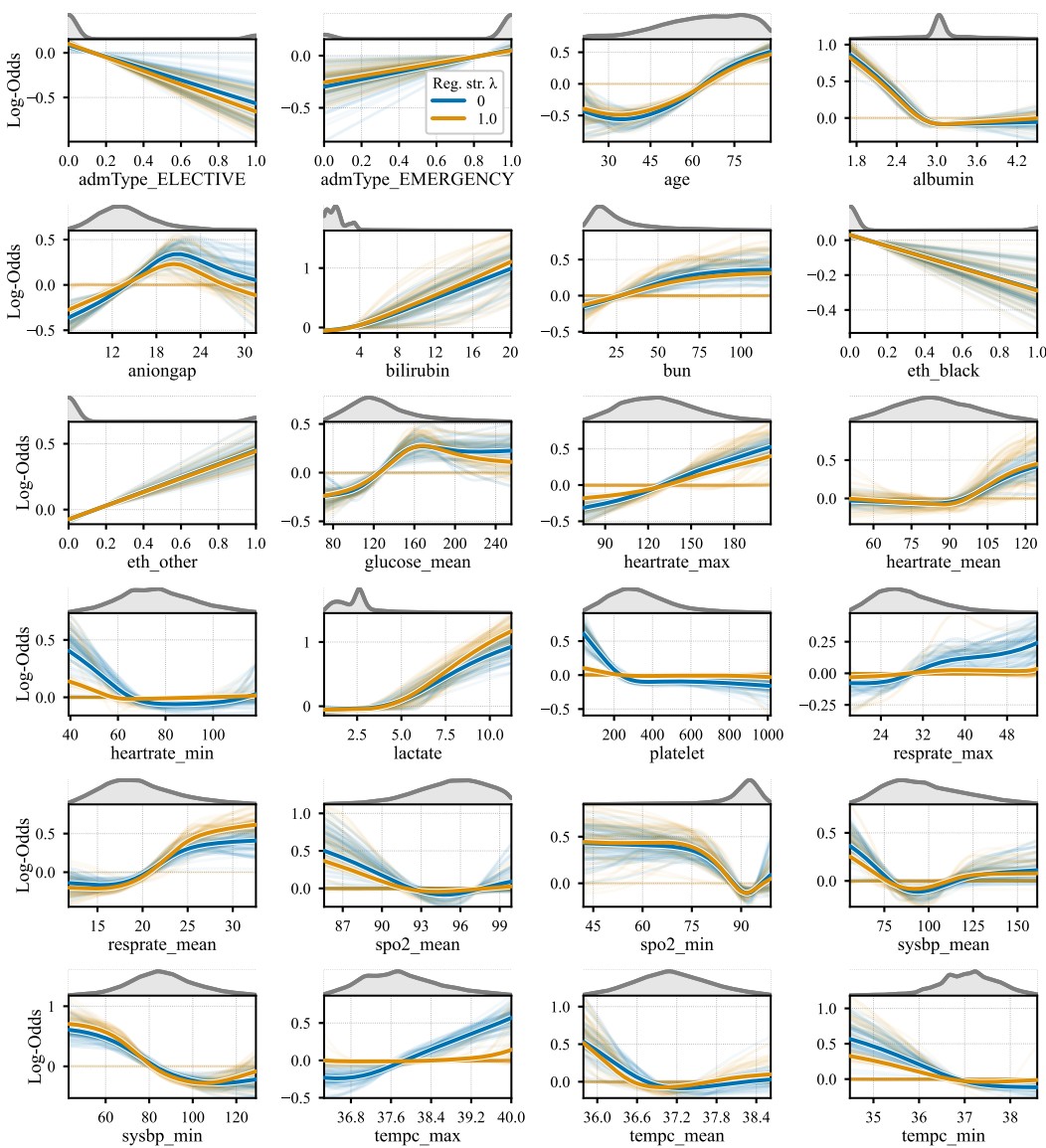

(a) Average and individual shape functions of the 24 out of the 57 features with the highest average importance score. A kernel density estimate of the training data distribution is depicted on top.

Figure 16: Results (part 2) for the **MIMIC-III dataset**. The considered NAMs were trained with and without concurvity regularization using 60 model initialization seeds each.

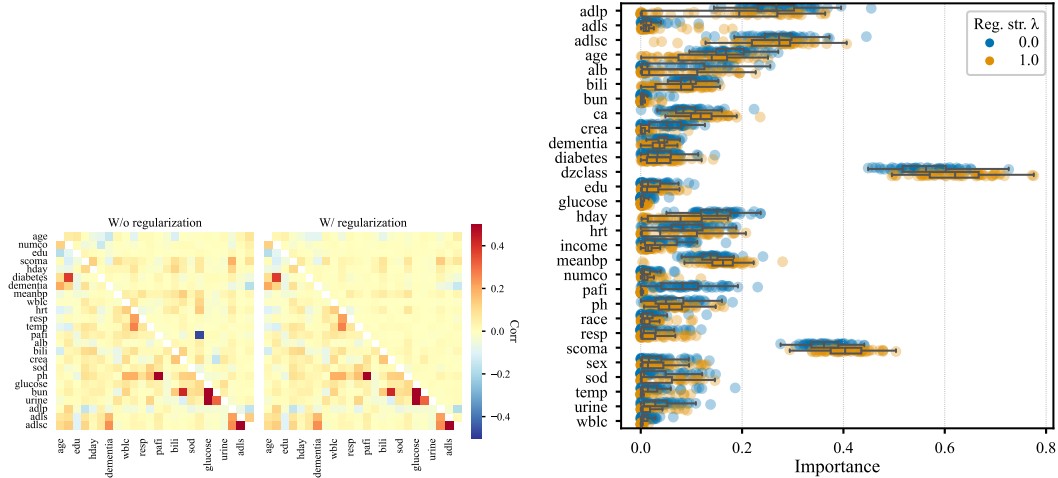

(a) Average feature correlations of the features (lower left) and non-linearly transformed features (upper right). Showing non-categorical features only.

(b) Combined box and strip plot of the models' feature importances.

Figure 17: Results (part 1) for the **Support2 dataset**. The considered NAMs were trained with and without concurvity regularization using 60 model initialization seeds each.

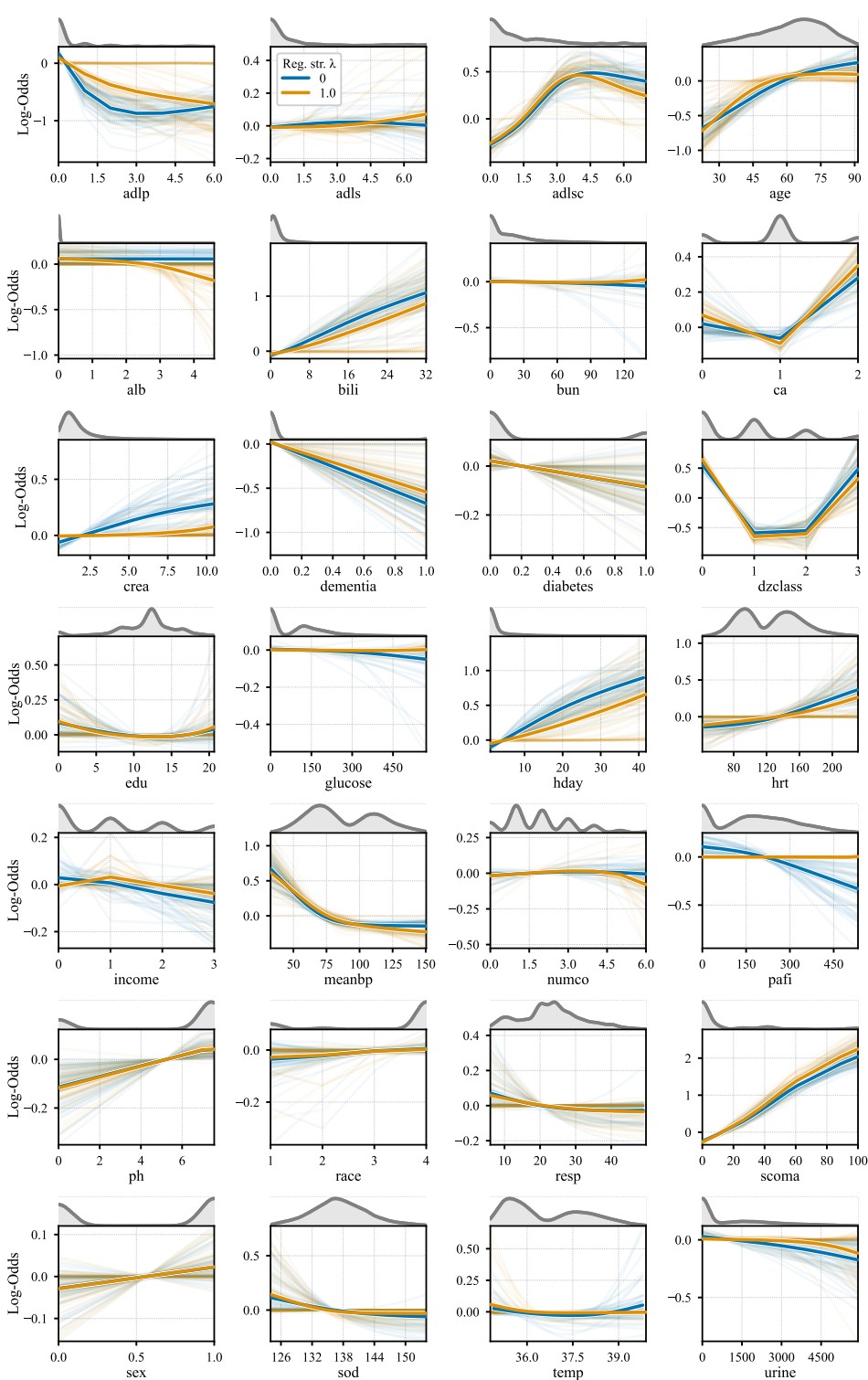

(a) Average and individual shape functions. A kernel density estimate of the training data distribution is depicted on top.

Figure 18: Results (part 2) for the **Support2 dataset**. The considered NAMs were trained with and without concurvity regularization using 60 model initialization seeds each.

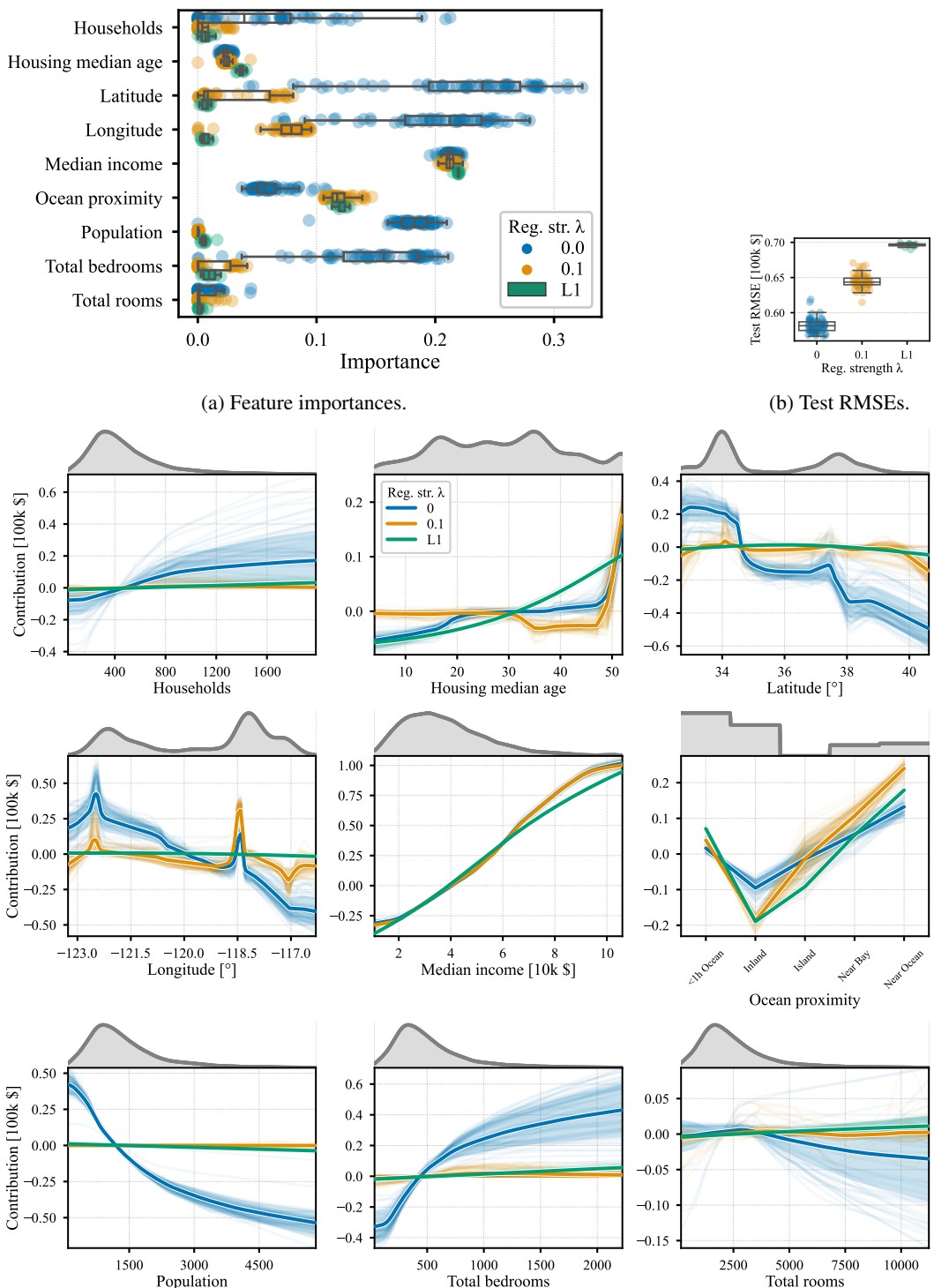

(a) Feature importances.

(b) Test RMSEs.

(c) Average and individual shape functions of the 9 selected features. A kernel density estimate of the training data distribution is depicted on top.

Figure 19: Comparison of concurvity and L1 regularization on the California Housing dataset. The considered NAMs were trained using 60 model initialization seeds each. Note that for L1, the regularization strength is set to $\lambda = 0.1$.

