# OpenReview forum: "Curve Your Enthusiasm: Concurvity Regularization in Differentiable Generalized Additive Models"
_NeurIPS.cc/2023/Conference — NeurIPS 2023 poster_

### Official Review · Reviewer_arAb · 2023-06-22

**Soundness:** 3 good
**Presentation:** 2 fair
**Contribution:** 3 good
**Rating:** 6
**Confidence:** 4

**Summary:**

The paper concerns the issue of concurvity in the context of Generalized Additive Models (GAMs), which can be considered as an extension of multicolinearity to GAMs. The authors propose a regularization scheme to reduce the concurvity between learned functions, hoping to improve interpretability of the model.

**Strengths:**

- Multicolinearity/concurvity are indeed  serious problems in statistics, and any method that can address such issues is of interest to the community.
- The proposed concurvity penalty is new and has not been discussed before.
- The numerical experiments show the promise of the proposed methodology.

**Weaknesses:**

- I think the numerical results although interesting, are rather limited. As far as I could tell, authors only discuss 4 tabular datasets, which is not enough. More datasets and baselines should be added---see [1] as an example.

[1] Chang, Chun-Hao, Rich Caruana, and Anna Goldenberg. "NODE-GAM: Neural Generalized Additive Model for Interpretable Deep Learning." International Conference on Learning Representations.

**Questions:**

- Connections to sparse regularization: As pointed out by the authors, the issue of multicolinearity can also arise in the context of linear models. It is well-known that regularization schemes such as sparse regularization can help to deal with correlation. For example, see [1,2]. The notion of variable selection in GAMs is also well-explored, see [3] and references therein. A natural question would be how such penalization schemes deal with concurvity. Intuitively, if one can reduce the total number of features used, a more compact model can be obtained that is less likely to suffer from concurvity.

- How should one choose the parameter $\lambda$? The authors mention the elbow method but I think this will need more exploration. It seems that the concurvity regularization can reduce validation accuracy. So, if one uses validation performance to choose a model, they'll end up with a model with no regularization. It is not clear to me how a practitioner can use the concurvity regularization if they have to sacrifice accuracy. Going back to the previous point, in the context of linear models, sparsity regularization usually comes with improved out-of-sample performance.

- Extensions: Another interesting direction is that GAMs might be too simple for more complex datasets, and methods that use higher-order interaction terms have been developed [3,4]. Would it be possible to extend the concurvity regularization to such interaction models? This definitely falls out of the focus of this paper, but a short discussion from the authors would be appreciated.




[1] Figueiredo, M., & Nowak, R.. (2016). Ordered Weighted L1 Regularized Regression with Strongly Correlated Covariates: Theoretical Aspects.

[2] Hazimeh, H., & Mazumder, R. (2020). Fast best subset selection: Coordinate descent and local combinatorial optimization algorithms. Operations Research, 68(5), 1517-1537.

[3] Chang, Chun-Hao, Rich Caruana, and Anna Goldenberg. "NODE-GAM: Neural Generalized Additive Model for Interpretable Deep Learning." International Conference on Learning Representations.

[4] Enouen, J., & Liu, Y. Sparse Interaction Additive Networks via Feature Interaction Detection and Sparse Selection. In Advances in Neural Information Processing Systems.

---

> ### Author Rebuttal · Authors · 2023-08-08
>
> Dear Reviewer arAb,
>
> Thank you for the thorough review of the paper, as well as for acknowledging the importance of the issue of concurvity and the novelty of our approach to mitigating this issue. We have addressed your concerns and questions as follows:
>
> **1. On the scope of evaluation:**
> We address this shared concern in more detail in our general comments C2 and C3. More specifically, as we argue, quantifying interpretability is yet an open question in the literature and a problem for the community to address. As such, we believe that simply evaluating accuracy and possibly sparsity or concurvity measures across a multitude of additional datasets provides limited additional insights.
>
> For these reasons we favored depth over width in our evaluation and chose a more detailed approach to assessing interpretability, illustrated in Figure 5 and, particularly, Fig. 6. We argue that understanding the impact on interpretability requires specific background knowledge on each dataset as showcased by our investigation of the California Housing dataset, limiting the value of evaluating a large benchmark.
>
> While quantifying interpretability is an important open question for the community, we argue that it is important to provide practitioners with tools to potentially deal with limited interpretability in their applications now, instead of waiting for progress on measuring interpretability.
>
> **2. On the connection to sparse regularization:**
> We agree with the reviewer that there is an interesting connection between our proposed concurvity regularization and sparse regularization and elaborate in more detail in general comment C2. However, a detailed comparison between these two regularization paradigms are outside the scope of our paper. For a thorough review of dealing with concurvity in spline-based GAMs via feature selection algorithms we refer the reviewer to Kovács 2022 [1].
>
> [1] László Kovács. Feature selection algorithms in generalized additive models under concurvity. Computational Statistics, pages 1–33, 2022.
>
> **3. On the choice of the regularization strength $\lambda$:**
> While we agree that model accuracy is paramount for a prediction model, there exist additional desiderata for interpretable models. Our concurvity regularizer enforces the requirement to eliminate self-canceling feature contributions thereby helping to avoid drawing false conclusions. While this leads to a slight decrease in model accuracy we feel that this is justified considering the increase in interpretability. Please see our global comment C1 for further elaboration of this argument. We propose to use the elbow method (or L-curve) to estimate the correct level of concurvity regularization but depending on the use-case, one could also choose e.g. a maximal 5% decrease in model performance to find the optimal trade-off. We further address this issue in our global comment C4.
>
> **4. On extending the approach to GAMs with higher-order interactions:**
> We thank the reviewer for the suggestion. Using concurvity regularization in higher-order interaction models would be straightforward, in the sense that any additional feature term can also be considered when calculating the pairwise correlation. However, it is currently unclear to us what kind of decomposition should be expected. We will be adding the following to the conclusion: “Moreover, it would be interesting to see how the concurvity regularizer works in differentiable GAMs that incorporate pairwise or higher-order interactions. Specifically, contrasting this with the ANOVA decomposition proposed by Lengerich et al. (2020) [2], in terms of single and pairwise interactions, could unveil some interesting insights.”
>
>
> [2] Lengerich, B., Tan, S., Chang, C., Hooker, G. &amp; Caruana, R.. (2020). Purifying Interaction Effects with the Functional ANOVA: An Efficient Algorithm for Recovering Identifiable Additive Models. Proceedings of the Twenty Third International Conference on Artificial Intelligence and Statistics, in Proceedings of Machine Learning Research 108:2402-2412
>
>
> Once again, we sincerely thank you for your constructive feedback and valuable insights. We look forward to the opportunity to further improve our work through continued dialogue.

---

> > ### Comment · Reviewer_arAb · 2023-08-10
> >
> > Thank you for clarifications and providing additional experiments. I think you rebuttal answers some questions, but I still think:
> >
> > - The paper would benefit from comparisons with sparse methods (or any other method that can reduce correlation).
> >
> > - Although the numerical experiments have improved, they are not 100% convincing.
> >
> > Based on these, I increase my score to weak accept, but I don't think a higher score would be fair.

---

> > > ### Author Response · Authors · 2023-08-11
> > > **Thank you for your swift response!**
> > >
> > > Dear Reviewer arAb,
> > >
> > > We thank you for your valuable feedback, and taking our responses into account. We happily acknowledge your willingness to upgrade your rating!
> > >
> > > We agree that including a comparison to sparsity regularization could benefit the paper, and are currently conducting further experiments to this end.
> > >
> > > Preliminary results indicate that L1 regularization on the feature contributions $f_i(x_i)$ can somewhat reduce concurvity, though a more substantial decreases in concurvity come at a much higher cost in increased validation error and overly aggressive feature selction. This is of course not surprising, given that we are now optimizing for a different measure. In the California Housing case study, we see that L1 regularization tends to select only very few features (which may vary depending on the random seed leading to some ambiguity in the interpretation of the model), whereas concurvity regularization assigns non-zero importance to several features, pruning mostly moderately to highly correlated features.
> > >
> > > This is in line with our previous statement that, unlike sparsity regularization, our proposed regularizer does not negatively affect features that do not show concurvity.
> > > We are happy to include results from these experiments in the camera-ready version.
> > > Finally, we note that there may be different motivations behind choosing sparsity regularization (as few features as possible) as compared to concurvity regularization (as decorrelated feature transformations as possible). If the goal is to remove concurvity, then sparsity regularization is a blunt tool, and vice versa. Thus, we believe these approaches to be complementary tools in building good, interpretable models.
> > >
> > > Sincerely,
> > >
> > > the Authors

---

### Official Review · Reviewer_aFW2 · 2023-07-06

**Soundness:** 2 fair
**Presentation:** 3 good
**Contribution:** 2 fair
**Rating:** 5
**Confidence:** 4

**Summary:**

The authors highlight concurvity as a relevant concern when developing additive models. They introduce a differentiable regularizer that aims to reduce concurvity (which will in general raise RMSE, though there may be a regularization strength that gives a good trade-off).

The proposed regularizer is applied to a neural additive model, which is demonstrated on three toy examples (two 2-variable examples where $Y$ equals one of the variables, one time series example with different weekly and daily step functions) and three UCI data sets (2x regression, 1x classification).


**Strengths:**

The paper provides a nice overview of concurvity and related works and why it is an important issue to be aware of. It introduces a novel regularizer that can easily be combined with any differentiable models.


**Weaknesses:**

The authors include a large set of references to related work on concurvity. Yet there is no comparison to other models (with the same proposed regularizer) or other methods of dealing with concurvity (despite mentioning multiple related approaches).
The paper claims that their concurvity regularization encourages feature selection (line 169) but there is no demonstration of this nor comparison to actual feature selection approaches.

Very limited evaluation on real-world data sets; see for example [b] in the same space which has an extensive comparison both on many data sets and across multiple methods. Cited work such as [25] (Kovács 2022) uses toy examples to much better effect; a comparison of the proposed NAM/regularizer approach on the same toy example would make for a much stronger submission.

Of course minimizing a given metric (such as the proposed concurvity regularizer) reduces that same metric, but I am missing a stronger demonstration of the practical benefits/relevance of this. The paper claims the resulting model is more interpretable, but does not really demonstrate this. See [a] for a discussion of interpretability of GAM features.

Though the discussion is all about concurvity and non-linearity, only scalar correlation values are shown in Fig. 6 (a). Pair plots of both $X_i$ vs $X_j$ and $f_i(X_i)$ vs $f_j(X_j)$ might demonstrate the effect of the regularization more easily. Claims such as “their feature contributions largely [cancel] each other out” (lines 282-284) could easily be shown by showing their sum across the data set.

### References:
- [a] *"How Interpretable and Trustworthy are GAMs?"* Chang, Tan, Lengerich, Goldenberg, Caruana (2021)
- [b] *"Additive Gaussian Processes Revisited"* Lu, Boukouvalas, Hensman (2022)


**Questions:**

### Substantial questions:

Q1. Time series with multiple seasonalities are in some sense a “degenerate” example as all functions depend on the same feature $t$. How, if at all, would time series have to be treated differently from “regular” data sets with multiple input features $X_1, \dots, X_p$?

Q2. Definition 2.2: you claim this “revised formal definition” as a main contribution of your work. How is it different from (and what are the similarities to) previous definitions?

Q3. Section 3: your description sounds like concurvity is purely a *model* issue, but is it not a property of the *data*?

Q4. Proof in Appendix A.1: this seems to hinge on your definition of correlation which assigns $\infty$ to the correlation with a constant vector. However, this correlation is simply ill-defined - and one could equally define it to be zero. Does this not invalidate your proof?

Q5. Your proposed regularizer scales quadratically with the number of additive components: how would you efficiently address this by parallelization? (Does the scaling not remain quadratically?)

Q6. “Our concurvity regularizer is agnostic to the function class …” (lines 140-142): “metrics proposed in the literature [42, 25] are not directly applicable.” [42] (Wood 2001) does not mention concurvity at all and while [25] (Kovács 2022) discusses concurvity, I could not find any mention of score/metric proposed in there. Moreover, what is the similarity of your proposed metric to that of [35] (Ramsay et al)?

Q7.a) Toy Example 2: Fig. 3 (a) suggests that without regularization, the model learns that $f_1(X_1) = 0$ *more* accurately (range of $f_1$ ~ 0.0015) than with concurvity regularization (range of $f_1$ ~ 0.02), how does this support the claimed benefit of regularization?
Q7.b) How do you see that $f_2$ approximates $|f_1|$ (lines 192-193)?

Q8. lines 238-241: could it be that this is due to it being classification rather than regression?

Q9. You discuss the distinction between multicollinearity and concurvity. Is it possible to have multicollinearity, but *not* have concurvity?

### Notation and clarity:

C10. Regarding the origin of concurvity (lines 100-101, 293), you might want to cite the work by Buja, Donnell, Stuetzle (1986) cited within your reference [9].

C11. The sum $\sum_{l=1}^N$ in the (GAM-Fit) and (GAM-Fit$_\perp$ equations seems to be inconsistent with the vector notation (the subscript $l$ is not used anywhere).

Q12. line 85: the notation used in the equation $\mathcal{H} \subset \dots$ is not immediately clear to me. Is $\mathcal{H}$ supposed to be a space of $p$-tuples?

Q13. “every suitable linear combination of features can be modified by adding a trivial linear combination” (lines 93-94) can you clarify what you mean here? Having gone through some of your references I understand, but your description on its own is rather confusing.
Likewise, “any non-trivial zero-combination of features can be added to a solution of (GAM-Fit)” (line 109-110): what does this mean?

C14. Simply writing “(GAM-Fit)” e.g. in line 110 made me take a while to realize that this is referring to an equation and to find where it was. How about having the equation label include a number, e.g. “(2; GAM-Fit)”, and then you can explicitly refer to “a solution of Eq. (2; GAM-Fit)” for example?

C15. Additional remarks in Appendix A.2: these are very helpful, would be great to at least summarize the points in the main text.

Q16. Figure 2 (b) is great to show that there is a value of $\lambda$ that reduces the concurvity measure without affecting RMSE perceptibly, but how sensitive is this to the value?
This might be easier to see in two separate plots of $\lambda$ vs $R_\perp$ and $\lambda$ vs RMSE overlaid on top of each other (to identify the range of $\lambda$ in which both are low simultaneously).

Q17. line 172: what is Fig. 8 supposed to demonstrate? This is not clear to me.

Q18. lines 175, 177: what would be the “scale” of $\lambda$? What does “moderate” or “considerably high” mean?

Q19. line 243: Are you aware of any other references that discuss the “elbow technique”? Thorndike’s “Who belongs in the family?” does not even mention “elbow”, and I would hope a better reference in this context is available (though it was a thoroughly enjoyable read which I thank the authors for bringing to my attention!)

C20. The references are inconsistent.

**Limitations:**

The authors acknowledge the limited validation of their approach (though I would not call it "diverse"). They also acknowledge that there is an interpretability-accuracy trade-off in concurvity regularization (though I find their demonstration of "increased interpretability" lacking).

---

> ### Author Rebuttal · Authors · 2023-08-09
>
> Dear Reviewer aFW2,
>
> We appreciate the time and effort you have invested in reviewing our submission. Your feedback is valuable and we are happy to clarify and expand on the main points raised in your review.
> Due to the character limit we were unable to include our answers to every question, but we will happily provide them upon additional request. The remaining questions are answered in our global comments.
>
> We have addressed your concerns as follows:
>
> **Weaknesses:**
> - Regarding the lack of comparison to other models and methods of dealing with concurvity:
> Our regularizer is only applicable to differentiable GAMs, with Neural Additive Models as the most prominent example. On the other hand, previous approaches dealing with concurvity investigate specific feature selection strategies (Kovács 2022), which are not straightforward to apply to NAMs (where all feature functions are fitted jointly). Since our work specifically focuses on differentiable GAMs, we have intentionally not included a direct comparison with previous concurvity reduction techniques. We are happy to point this out more clearly in the camera-ready version. Nevertheless, we agree that reporting a classical GAM baseline is useful. Therefore, we have added a spline-based GAM (via pyGAM (Servén et al. 2018)) as a baseline to our experiments. Please see also our global comment C3.
>
> - On the lack of demonstration of the claimed feature selection:
> Indeed, the concurvity regularizer encourages feature selection where features are strongly correlated, as demonstrated in Fig. 2a and 6. While feature selection is a possible consequence of concurvity regularization (when features are correlated) it is not the main aim and hence we do not compare with sparsity seeking methods, as detailed in our global comment C1.
>
> **Substantial questions:**
> 1. Indeed, we do not need to treat time series and regular tabular data differently in our approach. However, since time series form a data modality of independent interest, we have decided to put this example into a separate part. But we agree that it can be also considered a “degenerated” extreme case of our tabular data experiments.
> 2. Previous works do not give a precise definition. Our definition is loosely inspired by the one of [Ramsay et al., 2003], with an important difference: Ramsay et al. only consider model spaces of a cartesian form $\mathcal{H} = \mathcal{H}\_1 \times … \times \mathcal{H}\_p$, while we allow for any subset of $p$-tuples. This might appear like a minor detail but is the key to our insights in Sec. 3 and theoretically justifies why our regularizer is useful. In particular, spaces like $\mathcal{H}\_\perp$ would be incompatible with previous definitions.
> 3. Yes, one could say that concurvity is primarily a model issue. As we show, restricting the model space to  $\mathcal{H}\_\perp$ in Def. 2.2, concurvity can be ruled out, regardless of multicollinearity in the data. But of course, the data also plays an important role. For example, if the inputs $X_i$ are stochastically independent, so are any non-linear transformations. This implies that the regularization term is zero and does not affect the GAM-fit at all (see Remark (3) in Appendix A.2).
> 4. The assignment of infinity for constant features is just for convenience and we agree that a minor adaptation of the proof would be required for a different convention. This could be addressed by adding another constraint to $\mathcal{H}\_\perp$, excluding constant features. However, this technicality might cause confusion in the main body. Hence, we decided on the simplified version and address the ill-definedness of the correlation in Footnote 5.
> 5. We elaborate on this issue in general comment C5.
> 6. We agree that the paper introducing mgcv [42] (Wood 2001) does not mention concurvity, however, the concurvity indices are implemented in mgcv and are described in its manual. Hence, we find the reference appropriate. (Kovács 2022) has a good description of the concurvity indices implemented in mgcv on Page 7. Regarding the similarity of our metric compared to Ramsay et al.: They compute for each feature the correlation between $f_i$  and the sum of all other functions (excluding $f_i$), whereas we compute the pairwise correlation between all $f_i$ and $f_j$.
> 7. a.) We agree that the range of $f_1$ is closer to zero in the unregularized case; however, we also find that the unregularized model has introduced an almost perfect correlation between $f_1(X_1)$ and $f_2(X_2)$, which isn’t present in the data. Note that both models have the same validation RMSE. b.) From Figure 3 (a) one can see that $f_2(X_2)$ approximates $|f_1(X_1)|$ up to an affine transformation.
> 8. $R_\perp$ is measured in the (cartesian product) target space, so the regression target space or the space of raw logits in the case of classification. In both cases, the scale will depend on the target specifics and not on the target type. That is, generally, this does not result in $R\_\perp$ being on a much smaller scale in classification as opposed to regression tasks.
> 9. Yes,an example is the NeuralProphet experiment in Fig. 1, where time is used as input feature for all input components, implying perfect multicollinearity. Our approach demonstrates that the non-linearly transformed features can be decorrelated anyway. More generally, the idea of ruling out concurvity in the presence of (perfect) multicollinearity was precisely our motivation when deriving our regularizer in Section 3.
>
> Servén D., Brummitt C. (2018). pyGAM: Generalized Additive Models in Python. Zenodo. DOI: 10.5281/zenodo.1208723
>
> László Kovács. Feature selection algorithms in generalized additive models under concurvity. Computational Statistics, pages 1–33, 2022.
>
> T O Ramsay, R T Burnett, and D Krewski. The Effect of Concurvity in Generalized Additive Models Linking Mortality to Ambient Particulate Matter. Epidemiology, 14(1):18–23, 2003.
>
> We appreciate your feedback.

---

> > ### Author Response · Authors · 2023-08-10
> > **Minor notes on Notation and clarity**
> >
> > **Notation and clarity:**
> >
> > (10.) We thank the reviewer for the thorough analysis of the paper. To our understanding Buja, Donnell, Stuetzle (1986) is a preliminary technical report and not a reviewed paper. We were not able to find a digital version of (Buja 86) and hence decided to choose (Buja 89) which is widely cited as the seminal work on concurvity. The paper the reviewer is referring to appears to have later been published in 1994 as Donnell, Deborah J. et al. “Analysis of Additive Dependencies and Concurvities Using Smallest Additive Principal Components.” Annals of Statistics 22 (1994): 1635-1668.
> >
> > (11.) Thank you for spotting this inconsistency in our formulation of ERM. It slipped through and we have updated it accordingly.
> >
> > (12.) Yes, $\mathcal{H}$ can be any subset of $p$-tuples. But it is important to bear in mind that it is not necessarily a cartesian product of the form $\mathcal{H} = \mathcal{H}\_1 \times … \times \mathcal{H}\_p $, where $\mathcal{H}\_1 $, …, $\mathcal{H}\_p $ are individual function spaces.
> > The set $\mathcal{H}$ may impose additional constraints between the functions of a $p$-tuple, like in the definition of $\mathcal{H}\_{\perp}$. This relaxation might appear like a subtlety but is a crucial aspect in our definition of concurvity and derivation of our regularizer. See also Q2.
> >
> > (13.) We are happy to clarify this point in the camera-ready version. By “suitable linear combination” we basically mean a collection of coefficients fitting a target variable, say $Y \approx d_0 + \sum_i d_i * X_i$. In the presence of multicollinearity according to Def. 2.1, we would then have $Y \approx (c_0 + d_0) + \sum_i (c_i + d_i) * X_i$. So there exist other (infinitely many) equivalent solutions with completely different coefficients, which causes an undesirable ambiguity. The same argument applies to non-linear features in the case of concurvity in Def. 2.2.
> >
> > (14.) Good point, we will keep this in mind for the camera-ready version.
> >
> > (15.) Thanks! We agree and will expand the paragraphs where they are referenced in the main text.
> >
> > (16.) Thank you for the suggestion. We have added the suggested plot in Figure 3 of the rebuttal pdf. We also refer the reviewer to global comment C4.
> >
> > (19.) We agree that Thorndike’s “Who belongs in the family?” may seem counterintuitive to be the originator of the elbow method (or “L-curve”) as the term was only coined later. However, to the best of our knowledge, it is considered as such.

---

> > > ### Comment · Reviewer_aFW2 · 2023-08-14
> > > **Q17 and 18**
> > >
> > > Thank you for the additional explanations. Do you have answers for my questions 17 and 18 as well?

---

> > > > ### Author Response · Authors · 2023-08-14
> > > > **Commenting on Q17 and Q18**
> > > >
> > > > Yes, we originally left those out in our draft due to the character constraints and then they slipped through.
> > > >
> > > > **Q17 Regarding Fig 8 in the Appendix:**
> > > > This pairplot for Toy Example 1 showcases the decorrelation performance of our regularizer in the case of perfectly correlated inputs and can be seen as a more detailed Figure extending the content of Fig. 2 in the main text. Whereas the model trained without regularization shows strong (almost perfect) linear correlation of the outputs $f\_i(x\_i)$, our proposed regularizer effectively decorrelates said contributions. However, we acknowledge that some explanatory details are currently missing and we are happy to extend on this in the appendix.
> > > >
> > > > **Q18 Regarding the wording around the “scale” of $\lambda$:**
> > > > We believe that the wording “moderate” and “considerably high” regularization strength to become clear from context, in particular Fig 2b. Here, it can be observed, that regularization strengths of ca $10^{-3}$ lead to an eradication of concurvity without loss of accuracy. This is considered “moderate” given the scale of $\lambda$ in this experiment, ranging from $10^{-6}$ to $10^{1}$. In this context, regularization strengths of about $10^{0}$ to $10^{1}$ are considered “high”, given the mentioned range and the observed fact that it leads to a drastic decrease in predictive performance.

---

> > ### Comment · Reviewer_aFW2 · 2023-08-14
> >
> > Thank you for the overall thorough response to the comments from all reviewers. Overall, I will increase my score accordingly.
> >
> > Reading through the reviews & rebuttals again, I just wanted to get back to two comments from my review:
> >
> > ### more complex toy example
> >
> > > Cited work such as [25] (Kovács 2022) uses toy examples to much better effect; a comparison of the proposed NAM/regularizer approach on the same toy example would make for a much stronger submission.
> >
> > I would strongly encourage you to apply your method to Kovács's toy examples, whether for the camera-ready if accepted, or for a resubmission elsewhere if rejected - I believe this will make your paper significantly stronger: it would fill the gap between your current toy examples, which seem overly simplistic / unrealistic, and the evaluation on real-world datasets, where it's not clear what the answer ought to be as there is no ground truth available.
> >
> > If possible, if you can still run this and describe the results in comment, that would make it easier for me to finalise my opinion.
> >
> > ### pair plots of features and of transformed features
> >
> > > Though the discussion is all about concurvity and non-linearity, only scalar correlation values are shown in Fig. 6 (a). Pair plots of both Xᵢ vs Xⱼ and fᵢ(Xᵢ) vs fⱼ(Xⱼ) might demonstrate the effect of the regularization more easily. Claims such as “their feature contributions largely [cancel] each other out” (lines 282-284) could easily be shown by showing their sum across the data set.
> >
> > I would have appreciated being able to see these plots; I'm assuming you can't update the rebuttal PDF/add figures at this point, but again I think this is something that would make your work easier to understand and believe in.
> >
> > ### visualization of how to choose $\lambda$
> >
> > Thank you for preparing Fig. 3 of the rebuttal PDF - this does make it much easier to understand final choice of $\lambda$ on each dataset. For adding it to the manuscript, I would suggest also including vertical lines at the final choices of $\lambda$ for each column / dataset. E.g. based on these visuals I would expect the following choices:
> > - California Housing: $\lambda \approx 0.1$, at which point additional regularization no longer reduces concurvity
> > - Adult: slightly larger (maybe $\lambda \approx 0.2$ - harder to tell on a log plot without minor grid lines) until where the validation error is almost constant, while concurvity is continuously decreasing, and then the validation error suddenly starts increasing significantly
> > - Boston Housing: similar to Adult, a bit larger still (maybe $\lambda \approx 0.4$?).
> > Would be good to see indicated in the Figure whether this is indeed what you were thinking/choosing as well.

---

> > > ### Author Response · Authors · 2023-08-16
> > > **Additional results for the toy example by Kovács (2022)**
> > >
> > > **Regarding the toy example from Kovács (2022)**
> > >
> > > We thank the reviewer for taking our response into consideration and raising the evaluation score. In response to the suggestion to include a more complex toy example, we have replicated the toy example from Kovacs (2022) using our NAM setup. To recap, this example contains 7 features:
> > >
> > > $X_1 \sim X_2 \sim X_3 \sim U(0,1) $
> > >
> > > $X_4 = X_2^3 + X_3^2 + N(0, \sigma_1) $
> > >
> > > $X_5 = X_3^2 + N(0, \sigma_1) $
> > >
> > > $X_6 = X_2^2 + X_4^2 + N(0, \sigma_1) $
> > >
> > > $X_7 = X_1 \times X_2 + N(0, \sigma_1)$
> > >
> > > $Y = 2X_1^2 + X_5^3 + 2 \sin (X_6) + N(0, \sigma_2)$
> > >
> > > where $\sigma_1$ is sufficiently small to create severe concurvity among the features ($\sigma_1 = 0.05$, $\sigma_2 = 0.5$).
> > > We simulated 10,000 data points from this model and created a 7:3 train/test split. We fitted 20 random initializations of a NAM in unregularized, concurvity regularized, and L1 regularized settings. The regularization parameter $\lambda$ was determined separately for each regularization type based on trade-off curves. For concurvity regularization we used $\lambda = 0.1$ and for L1 we used $\lambda = 0.05$.
> > >
> > > The results, reported as the $R^2$ on the test set, are reported in the table below, with the top three rows from Kovács (2022) for comparison. Confidence intervals of the mean are estimated on 10000 bootstrap samples. Features are presented in descending order of their importance (as defined in the main paper) for the best-fitting model of each setting, with importances reported in the cell below.
> > >
> > > | **Model** | **Selected Features** | **$R^2$(test) (%)** mean, (5% / 95% conf. int) | **$R_\perp$(test)** mean, (5% / 95% conf. int) |
> > > |---|---|---|---|
> > > | Full model | Full model | 84.99 |  |
> > > | Stepwise | **X1**, X4, **X5**, **X6** | 85.11 |  |
> > > | Hybrid algorithm | **X1**, **X5**, **X6** | 85.31 |  |
> > > | Unregularized (ours) | **X1**, **X6**, X4, X2, **X5**, X3, X7 | 80.77, (80.31 / 80.95) | 0.22, (0.20 / 0.23) |
> > > |  ⤷ Feature Importance | **0.129**, **0.097**, 0.066, 0.053, **0.043**, 0.013, 0.004 |  |  |
> > > | Concurvity Reg. (ours) | **X1**, **X6**, **X5**, X2, X7, X4, X3 | 79.28, (78.52 / 79.88) | 0.03, (0.02 / 0.03) |
> > > |  ⤷ Feature Importance | **0.132**, **0.125**, **0.088**, 0.070 , 0.006, 0.002, 0.002 |  |  |
> > > | L1 Reg. (ours) | **X6**, **X1**, **X5**, X7, X4, X3, X2 | 79.12, (78.50 / 79.47) | 0.21, (0.20 / 0.21) |
> > > |  ⤷ Feature Importance | **0.147**, **0.106**, **0.037**, 0.009, 0.008, 0.005, 0.0  |  |  |
> > >
> > > We note that both concurvity regularization and L1 regularization correctly identifies the three predictive features $X_1$, $X_5$ and $X_6$ on which $Y$ directly depends. This is not the case without regularization. Furthermore, we find that concurvity regularization effectively reduces $R_\perp$, unlike L1 regularization.
> > >
> > > We also note that the $R^2$ values of all NAM implementations are lower than those reported by Kovacs, which we believe is due to the inductive biases in spline-based models being particularly well-suited to the mostly polynomial-based problem.
> > > We appreciate the reviewer's insightful suggestion, which has indeed underscored the efficacy of concurvity regularization. We are more than willing to provide further results for the toy example, should the reviewer have any specific requests. We would be pleased to incorporate these results into the final version of the paper.
> > >
> > >
> > > **Regarding the pair plots**
> > >
> > > Regrettably, we are unable to update the rebuttal PDF at this point. Initially, we opted not to include the pair plots due to space limitations. Upon further examination, we found that these plots provided minimal extra insights compared to the scalar correlation plot, while taking up significantly more space. Consequently, we believe they are more suited to the appendix rather than the main body of the paper. We would be glad to incorporate them into the final version of the paper.
> > >
> > > **Visualization of how to choose** $\lambda$
> > >
> > > We thank the reviewer for the suggestion and will include the suggested plots with vertical lines indicating the chosen regularization strength in the final version of the paper.

---

### Official Review · Reviewer_17dg · 2023-07-07

**Soundness:** 3 good
**Presentation:** 2 fair
**Contribution:** 3 good
**Rating:** 6
**Confidence:** 3

**Summary:**

Generalized additive models (GAMs) offer greater interpretability than other machine learning methods while having greater flexibility than generalized linear models. This paper addresses how to account for concurvity (the non-linear analog to multicollinearity in linear regression). The authors suggest directly penalizing correlations between the features during training. Concurivity poses problems with interpretability (which (transformed) feature to attribute the contributions to) and can increase the variance in the model. The proposed regularizer was found to reduce concurvity while maintaining strong predictive performance. The viability and success of this method are shown in time series forecasting as well as tabular regression tasks.

**Strengths:**

- Clear motivation for the problem and general organization of the paper
- Method is light on assumptions regarding the feature transformations, $f_i$ so this can be applied in many settings
- Improves model interpretability while reducing variance in fitted functions

**Weaknesses:**

- The figures were very information dense (particularly fig 6). More discussion of these results and how to interpret them would aid the reader
- Not much space was spent on the time-series experiments relative to how much was dedicated to it in the introduction


Minor
- line 75: (linear) regression
- line 83: minimize over $\beta$ as well?
- line 126: minimize over $\beta$?



**Questions:**

- How much do the $f_i$ differ depending on $\lambda$? How fair is it to compare feature importances across different levels of regularization if the resulting $f_i$ are not the same? (Possibly addressed in figure 6.c?)

- line 126.5: in the $gam-fit_\perp$ what is the $l$ index for? It does not show up in any of the other terms

- What is the additional computational burden of computing the regularizer? How well does this scale?

- How does gam-fit perform in the presence of covariates that are neither perfectly correlated nor entirely uncorrelated (as they were in the toy examples)?

---

> ### Author Rebuttal · Authors · 2023-08-08
>
> Dear Reviewer 17dg,
>
> Thank you for your constructive feedback and recognition of the clear motivation for our problem, the broad applicability of our method, and its contribution to improving model interpretability and reducing variance in fitted functions. We have addressed your concerns and questions as follows:
>
> 1. On the information density of the figures:
> We agree that, given the space constraints of the submission, some figures contain quite a lot of insights. We happily extend our discussion in the camera-ready version.
>
> 2. On the space dedicated to time-series experiments:
> The main reason why we did not devote more space to time-series experiments is their “degenerated” nature in the context of GAMs. Indeed, in the NeuralProphet example, time is used as the only input feature for all additive components, so that perfect multicollinearity is present. While our approach can be very useful in such scenarios, it should be considered an extreme case of our tabular data experiments, where the input feature relationships are more complex.
> On the other hand, the special form of the time-series example in Fig. 1 makes it intuitive and therefore well-suited for an introduction to our approach without prior knowledge of concurvity. Hopefully, this clarifies the mismatch in dedicated space. We are happy to point out this aspect in the camera-ready version.
>
> 3. On the minor errors:
> We appreciate your attention to detail and have adjusted the paper accordingly.
> 4. On the difference in $f_i$ depending on $\lambda$:
> We fully agree that this is tricky in general. As also addressed in our global comment C2, quantifying increased interpretability is challenging and there exists no gold standard yet. We choose to compare the shape functions $f_i$ and aggregated feature importances of NAMs trained with and without concurvity regularization in our case study. After all, the shape functions describe the prediction mechanism of GAMs and are used to gain insights or even guide practitioners in making decisions alongside the aggregated feature importances. The main purpose of Fig. 6(b) is therefore not a direct quantitative comparison of feature importantes between different regularization levels, but it should be rather used to extract qualitative insights, such as larger variances or bi-modality. We are happy to clarify this in the camera-ready version.
> 5. On the additional computational burden of computing the regularizer:
> We address this in our global remark C5. We will clarify the computational scaling in more detail in the camera-ready paper.
> 6. On the performance of gam-fit in the presence of covariates that are neither perfectly correlated nor entirely uncorrelated:
> We thank the reviewer for suggesting this experiment which was similarly suggested by reviewer 5zAd. We added an additional result for toy example 1 in Figure 4 of the rebuttal pdf (center row) where the features have a correlation of 0.9. We find that w/o regularization the model converges to the wrong solution in every case while w/ regularization the model converges to the right solution in almost all cases.
>
> We hope that these responses address your concerns and we are open to further discussions. We appreciate your feedback and will make the necessary adjustments in the camera-ready version of the paper.

---

> > ### Comment · Reviewer_17dg · 2023-08-14
> >
> > Thank you for the clarifying comments. I will update my score to weak accept.

---

> > > ### Author Response · Authors · 2023-08-15
> > >
> > > Thank you Reviewer 17dg for raising your score. We are happy to clarify any remaining concerns.

---

### Official Review · Reviewer_5zAd · 2023-07-07

**Soundness:** 3 good
**Presentation:** 4 excellent
**Contribution:** 2 fair
**Rating:** 4
**Confidence:** 5

**Summary:**

The paper proposes a regularization using pairwise correlations of shape functions for differentiable GAM, aimed at reducing concurvity between shape functions

**Strengths:**

1.	The idea of the paper is well-explained and straightforward.  The correlated terms can be self-cancelled to avoid to unnecessary complexity. So reducing concurvity may potentially enhance generalizability.
2.	The paper’s story, theory and the numerical experiments all demonstrate that regularization effectively controls the correlations of the non-linearly transformed features.
3.	The proposed component is easy to implement and optimize and as far as I understand, it can be incorporated into any differentiable GAM.


**Weaknesses:**

1.	The foundation of this paper – the reason why correlated terms should be cancelled out is still beyond me. For the time series scenario in the introduction, I understand that high frequency terms of feature pairs hinder interpretation. But what if the correlation does lie in the ground truth? Especially for the tabular data, the visualized results in the public dataset show that most of the features have almost no prediction power for the target variable (e.g. Population, households, housing age before 30 years, total bedrooms and total rooms, etc.), which seem to be counterintuitive. Little concurvity is allowed, but it happens in the real world. Is it possible that restricting concurvity may in general push the shape functions away from their natural relationship (trained to be not effective but actually affects the target)? The author may provide further evidence and explanation regarding the negative effects of concurvity, to make the article more convincing.
2.	While previous literature found that regularizing the cross-covariance increases generalization performance, the numerical results presented in this paper indicate that controlling the concurvity will sacrifice the generalization ability anyhow. The accuracy is slightly worse than the one without regularization, in exchange of the stated ‘interpretability’, which unfortunately is not straightforward in the figures. Therefore, it appears that the trade-off between accuracy and correlation may not be worth it in this case.
3.	The paper showcases the motivation using a time-series example, but no visual results for time series are presented to address the issue. It is doubtful whether the regularization does control the frequency or some uninterpretable parts, rather than merely force terms to be uncorrelated (in an unnecessary manner).
4.	It is not shown how to do the optimization. Is the calculation of regularization time-consuming and unstable? What would happen if the regularization is implemented in every iteration rather than only 5% of the total optimization steps as introduced in the article? It seems a mismatch between the design and the realization.

**Questions:**

1.	In Figure 2(a), we see the model with regularization chooses a random point on a curve. This is because both features are identical in the setting, making the model unidentifiable. What if the two features are correlated but not perfectly correlated? Is the regularizer able to identify the true predictive variable X1, and not so affected by random seeds?
2.	How to determine a proper level of the strength of regularization (lambda)?

**Limitations:**

yes

---

> ### Author Rebuttal · Authors · 2023-08-08
>
> Dear Reviewer 5zAd,
>
> Thank you for your comprehensive review and constructive feedback on our paper. We appreciate your recognition of the clarity of our idea, the effectiveness of our regularization in controlling the correlations of the non-linearly transformed features, and the ease of implementation and optimization of our proposed component.
>
> We have carefully considered your concerns and questions and have addressed them as follows:
>
> **Weaknesses:**
> 1. On the concern about the foundation of the paper and the negative effects of concurvity: Thank you for you for this insightful comment. Since related points were brought up by the other reviewers as well, we have added a global statement; please see comment C1. Regarding your specific concern: While we agree that ground-truth input features often exhibit natural relationships due to correlation, we do not think that the behavior of our regularization approach is counterintuitive. In fact, the idea of “suppressing” redundant features is well-established in the field of feature selection and widely-accepted in the community. These modes intend to provide a particularly simple predictive model, relying on as few input features as possible. However, this does not mean that the other features have no predictive power (they almost always have). Our approach aligns well with this approach, as it provides a particularly simple model by decorrelation the target features.
> Having said this, we agree that it is useful to provide more evidence that ignoring concurvity can be very problematic. To this end, we have added a spline-based GAM (via pyGAM (Servén et al. 2018)) as a baseline to our experiments (see also global comment C3), which demonstrates that the resulting shape functions exhibit a large variance – an issue commonly reported in the concurvity literature (please see the review pdf). In other words, the shape functions do not exhibit a natural relationship, since the space of shape functions is degenerate with possibly infinitely many equivalent solutions. We demonstrate that our method can greatly reduce this ambiguity, and in that sense, make the prediction models more interpretable.
> We hope that this clarifies your concern about the foundation of our approach and we are happy to make this point clearer in the camera-ready version.
>
> Servén D., Brummitt C. (2018). pyGAM: Generalized Additive Models in Python. Zenodo. DOI: 10.5281/zenodo.1208723
>
> 2. On the trade-off between accuracy and correlation:
> We are assuming that you are referring to the work by Cogswell 2016 by “regularizing the cross-covariance increases generalization performance”. While the work by Cogswell et al. indeed proposes a decorrelation approach, it operates in a fairly different regime, namely decorrelating _hidden representations_ to reduce overfitting of deep neural networks.  Although regularization often improves generalization of deep neural networks by reducing overfitting, we did not expect the same improvement in generalization in GAMs since overfitting is typically not a major issue in this type of models. Rather, regularization is employed here to constrain the properties of the model with the aim of improving interpretability.  The trade-off between accuracy and concurvity mirrors the well-established sparsity-accuracy trade-off in classical feature selection paradigms, and our experiments demonstrate that the sacrifice in accuracy is fairly small in all considered cases. We are not aware of any previous works where regularization has improved generalization in GAMs; if the reviewer has any such works in mind we would be happy for a pointer to these references. Regarding the gain of interpretability, we kindly refer to our answer to your first concern as well as global comment C1.
>
> M Cogswell, et al. “Reducing Overfitting in Deep Networks by Decorrelating Representations.” In: International Conference on Learning Representations, 2016.
>
> 3. On the lack of visual results for time series:
> We believe this to be a misunderstanding: the example we present in Figure 1 shows actual experimental results obtained for the three settings detailed in Section 4.2. The only learnable parameters of the model are the coefficients of the Fourier terms of which each has a distinct frequency. As a result, the decorrelation does directly influence the dominance of a frequency since the model has no other parameters to adapt.
>
> 4. On the optimization process:
> We thank the reviewer for raising this concern. We elaborate more on the efficiency and scaling of the regularization in the general comment C5. We want to clarify that the regularization is added after a warm-up phase of 5% of the total optimization steps, but afterwards used *in every optimization step*. We will clarify this in Appendix C.1.
>
> **Questions:**
> 1. On the questions about Figure 2(a) and what happens if two features are correlated but not perfectly correlated.
> Thank you for suggesting this experiment which was similarly suggested by reviewer 17dg. We added an additional result for toy example 1 in Figure 4 of the rebuttal pdf (center row) where the features have a correlation of 0.9. We find that w/o regularization the model converges to the wrong solution in every case while w/ regularization the model converges to the right solution in almost all cases.
>
> 2. How to determine a proper level of the strength of regularization (lambda)? We address this in our global comment C4.
>
>
> We hope that these responses satisfactorily address your concerns and provide further clarification on our work. We are open to continuing this dialogue to further refine our paper. We truly appreciate your insightful feedback and will incorporate your suggestions in the final version of our paper.

---

> > ### Author Response · Authors · 2023-08-21
> >
> > We would like to thank the reviewer again for the thorough review and hope we have addressed your concerns.
> >
> > We would appreciate it if you could consider adjusting your score accordingly.

---

### Author Rebuttal · Authors · 2023-08-08

We thank all reviewers for carefully reading our manuscript as well as their thoughtful comments and suggestions. We are happy that the reviewers acknowledge that “multicollinearity/concurvity are indeed serious problems in statistics” (R.arAb) and that our approach is novel (R.arAb), light on assumptions (R.12dg), easy to implement and optimize (R.5zAd), can be combined with any differentiable GAM (R.aFW2 & R.17dg), and that it improves the model interpretability (R17dg). Thus, we believe our contribution to be of value to the NeurIPS community.

We summarize and address some questions shared amongst the reviewers in the following.

### C1. Motivation of our contribution and clarifications on concurvity in general
While all reviewers agree on the efficacy of our regularizer, there were some questions as to the motivation to reduce concurvity, which we address here.

Although concurvity may be a property of datasets (as illustrated in our two toy examples), we argue that it is an undesirable property of a model. Akin to linear models in the presence of multicollinearity, concurvity in GAMs produces a large variance in model fits (see Fig. 2c att. pdf) and may introduce spurious correlations between the transformed features. Both phenomena may be observed in the California Housing case study. In particular, some features with positive correlation in the input space (e.g., “Total bedrooms” and “Population”) become negatively correlated in the transformed feature space of the unregularized NAM, leading to canceling contributions. In conventional statistics, it is not uncommon to drop features or employ sparsity regularization in order to remove multicollinearity (Dormann et al., 2013). However, sparsity regularization penalizes large vector norms of transformed features, thus also affecting model fit in the absence of concurvity. Thus, if the primary goal is to remove spurious correlations from a model rather than to reduce the number of features, concurvity regularization may be preferable.

Restricting concurvity reduces variance at the expense of increasing bias, generally resulting in a reduced accuracy. As we do not regularize in order to reduce overfitting, we did not expect improved generalization from regularization.

### C2. On quantifying interpretability
We note that quantifying interpretability is an unsolved issue in the community. While Chang et al. [a] attempt to quantify interpretability using sparsity and fidelity metrics, these cannot be considered gold standard or best-practice as they pose additional concerns and limitations. Whereas data fidelity can only be measured on synthetic datasets, sparsity “can hide data bias and discriminate against minority groups.” [a].

Therefore, we chose a more detailed approach to assessing interpretability, illustrated in figures 5 and 6 and the detailed discussion of the California Housing case study. Here, we note that regularization mitigates inflated feature importances in correlated features, removes spurious correlations, reduces variance in the shape functions, and prunes some correlated features while leaving uncorrelated features intact.

While quantitative measures of interpretability are still lacking, we argue that practical tools to deal with this issue in the meantime are still needed. We believe our contribution serves as an alternative to sparsity-based regularization, thus contributing to the toolkit of building interpretable models.

[a] C Chang, S Tan, B Lengerich, A Goldenberg, and R Caruana. “How Interpretable and Trustworthy are GAMs?” In ACM SIGKDD Conference on Knowledge Discovery & Data Mining, p 95–105, 2021.

### C3. On the evaluation of our regularizer
A concern shared between most reviewers is the scope of our evaluation, currently comprising three toy examples  and four real-world tabular datasets. In response to these concerns, we have conducted additional experiments on three additional tabular datasets. Moreover, we have added evaluation of a conventional spline-based GAM across all tabular datasets, which is depicted in the attached PDF (Fig. 1&2c) and will be included in our camera-ready version. The results are clearly in line with our previous findings and further demonstrate the applicability of our method.

Finally, we will provide further evaluation of the investigated datasets, similar to our in-depth analysis presented in Fig. 6, in the appendix of the camera-ready version.

### C4. On regularization strength lambda in practice
As some reviewers pointed out, choosing the regularization strength lambda is of great practical importance. We briefly mention the elbow technique (or L-curve) in our paper, using the provided tradeoff curves (e.g. Fig 2b, 3b, 4 and 5). While these curves are well suited to identify a good tradeoff point between gains (in terms of lower concurvity) and losses (in terms of less accuracy), identifying the corresponding lambda is arguably tricky in the printed version (but trivial in an interactive digital plot). Alternatively, we now also provide separate curves for concurvity and accuracy over lambda, as suggested by R.aFW2 – see the attached PDF.

### C5. On computational complexity and overhead
A last general concern among reviewers was computational overhead, given the quadratic scaling of the regularizer in the number of features mentioned in the paper. An important point here is that the calculation of the pairwise correlations can be parallelized via vectorization, making it efficient to calculate while keeping the scaling constant controllable (i.e. via increased parallel compute). As an example, even for a dataset of around 1000 columns/features (which is way beyond most typical datasets) at a batch size of 512 our implementation of the proposed concurvity regularizer has a negligible average runtime of 6.9 ms (tested on an M1 MacBook Pro averaged over 1000 runs). We will provide a small analysis of the runtime and overhead in the appendix of our revised paper.

---

### Decision · Program_Chairs · 2023-09-21

**Decision:**

Accept (poster)

**Comment:**

The reviewers praise the important problem, original idea, writing of the paper, ease of implementation, improved interpretability. The main criticisms are the somewhat unclear motivation for the method, tradeoff between performance and interpretability, computational considerations, choice of hyperparameters, limited scope of experiments, and lack of baselines. However, these concerns have been mostly addressed by the rebuttal.